# Agonistic CD40 therapy induces tertiary lymphoid structures but impairs responses to checkpoint blockade in glioma

Luuk van Hooren [1,11], Alessandra Vaccaro[1,11], Mohanraj Ramachandran[1], Konstantinos Vazaios[1], Sylwia Libard[1,2], Tiarne van de Walle[1], Maria Georganaki[1], Hua Huang[1], Ilkka Pietilä[1], Joey Lau [3], Maria H. Ulvmar[1], Mikael C. I. Karlsson [4], Maria Zetterling[5], Sara M. Mangsbo[6], Asgeir S. Jakola[7,8], Thomas Olsson Bontell[9,10], Anja Smits[5,8], Magnus Essand [1] & Anna Dimberg [1✉]

Gliomas are brain tumors characterized by an immunosuppressive microenvironment. Immunostimulatory agonistic CD40 antibodies (αCD40) are in clinical development for solid tumors, but are yet to be evaluated for glioma. Here, we demonstrate that systemic delivery of αCD40 in preclinical glioma models induces the formation of tertiary lymphoid structures (TLS) in proximity of meningeal tissue. In treatment-naïve glioma patients, the presence of TLS correlates with increased T cell infiltration. However, systemic delivery of αCD40 induces hypofunctional T cells and impairs the response to immune checkpoint inhibitors in pre-clinical glioma models. This is associated with a systemic induction of suppressive CD11b+ B cells post-αCD40 treatment, which accumulate in the tumor microenvironment. Our work unveils the pleiotropic effects of αCD40 therapy in glioma and reveals that immunotherapies can modulate TLS formation in the brain, opening up for future opportunities to regulate the immune response.

[1] Department of Immunology, Genetics and Pathology, Science for Life Laboratory, The Rudbeck Laboratory, Uppsala University, Uppsala, Sweden. [2] Department of Pathology, Uppsala University Hospital, Uppsala, Sweden. [3] Department of Medical Cell Biology, Uppsala University, Uppsala, Sweden. [4] Department of Microbiology, Tumor and Cell Biology, Karolinska Institutet, Stockholm, Sweden. [5] Department of Neuroscience, Neurology, Uppsala University, Uppsala, Sweden. [6] Department of Pharmaceutical Biosciences, Science for Life Laboratory, Uppsala University, Uppsala, Sweden. [7] Department of Neurosurgery, Sahlgrenska University Hospital, Gothenburg, Sweden. [8] Department of Clinical Neuroscience, Institute of Neuroscience and Physiology, Sahlgrenska Academy, University of Gothenburg, Gothenburg, Sweden. [9] Department of Physiology, Institute of Neuroscience and Physiology, Sahlgrenska Academy, University of Gothenburg, Gothenburg, Sweden. [10] Department of Clinical Pathology and Cytology, Sahlgrenska University Hospital, Gothenburg, Sweden. [11] These authors contributed equally: Luuk van Hooren, Alessandra Vaccaro. ✉email: anna.dimberg@igp.uu.se

Glioblastoma (GBM), or grade IV glioma, is the most common malignant primary brain tumor in adults. Despite multimodal treatment strategies with surgery, radiotherapy, chemotherapy, and recently tumor-treating fields, the outcome for GBM patients remains poor with a median survival of less than 24 months[1]. While checkpoint inhibitors (CPIs) targeting PD-1 and CTLA-4 have seen clinical success in several human solid tumors[2] and experimental murine glioma models[3], their efficacy has proven limited in GBM patients[4]. This is not surprising since brain immune responses are adapted to a sensitive and immunospecialized microenvironment. Therapeutic approaches designed specifically for the brain tumor micro-environment are therefore needed to mount an effective immune response against GBM.

Gliomas are highly infiltrated by bone marrow-derived macrophages and brain-resident microglia, which promote tumor growth and suppress the immune response[5]. CD40 is expressed on several antigen presenting cells (APCs) and agonistic CD40 antibodies (αCD40) have broad immunostimulatory effects. Indeed, αCD40 can polarize macrophages towards a tumor-suppressive profile and enhance antigen presentation by dendritic cells (DCs)[6–8]. Moreover, CD40 activation of B cells regulates activation, antibody production, germinal center formation and antigen presentation[9,10]. αCD40 antibodies are currently in clinical development for numerous solid tumors[11]. However, there is conflicting evidence regarding their efficacy in glioma since outcomes have varied depending on experimental model and combinatorial treatment regimen[12–14]. A thorough understanding of how αCD40 therapy impacts different compartments of the brain immune response is necessary to evaluate its potential for the treatment of glioma.

Tertiary lymphoid structures (TLS) are ectopic lymphoid aggregates that form at sites of chronic inflammation and resemble secondary lymphoid organs, including a B cell follicle surrounded by a T cell zone with mature DCs and a network of follicular dendritic cells (FDCs)[15,16]. Ectopic expression of lymphotoxin (LT) is instrumental in driving TLS formation[17,18]. The engagement of the $LT\alpha_1\beta_2$ heterotrimer (or the homologous ligand TNFSF14) with the lymphotoxin β receptor (LTβR) expressed on stromal cells induces chemokines associated with leukocyte recruitment[19], the formation of FDC networks and of germinal centers in B cell follicles[20]. Importantly, TLS affect disease progression. TLS are considered to aggravate the inflammatory response in autoimmune disease such as multiple sclerosis (MS)[21], while in several types of cancer they are associated with improved response to immunotherapy and a favorable prognosis[22–26]. B cells are the dominant component of TLS and B cell-depleting antibodies recently gained clinical approval for MS patients as they ameliorate disease severity[27], while in melanoma depletion of B cells reduces CD8$^+$ T cell infiltration[28]. In the context of cancer, TLS are believed to provide an alternative to tumor-draining lymph nodes as a site of antigen presentation and activation of naïve T cells[16]. Notably, the presence of TLS in glioma has not been described to date.

In this study, we demonstrate that systemic exposure to αCD40 in glioma-bearing mice leads to reduced CD8$^+$ T cell cytotoxicity and impairs the response to CPIs. This is associated with the expansion of a suppressive CD11b$^+$ B cell population. However, αCD40 stimulation of B cells also enhances the formation of TLS in the brain of glioma-bearing mice. Interestingly, TLS are present in human glioma and correlate with increased intratumoral T cells in GBM, suggesting an association between TLS and regulation of immune responses in glioma patients. Our study demonstrates the presence of TLS in human glioma and reveals the multifaceted effects of αCD40 in murine glioma models, which have important therapeutic implications.

## Results

**αCD40 enhanced TLS formation in murine glioma models**. To investigate the effects of agonistic CD40 antibodies (αCD40) on the tumor microenvironment, we intravenously administered αCD40 or the corresponding rIgG2a isotype control to C57BL/6 mice with syngeneic gliomas. Immunofluorescence staining of brain sections from glioma-bearing mice revealed the presence of immune cell clusters with a distinct core of B cells that were reminiscent of TLS (Supplementary Fig. 1a and Supplementary Movie 1). We defined TLS as compact clusters of CD45$^+$ cells with a dense core of B220$^+$ B cells. αCD40 was associated with increased numbers and total surface area of TLS in both GL261 (Fig. 1a–c) and CT-2A (Fig. 1d–f) glioma models. The TLS were consistently located close to the meninges (around the cortex or close to choroid plexuses) in proximity of the tumor tissue (Supplementary Fig. 1a, b). TLS did not form in the brain of untreated tumor-free mice or after mock injection of tumor cells followed by αCD40 therapy (Supplementary Fig. 1c). In summary, TLS were observed in brains of glioma-bearing mice and αCD40 treatment enhanced their formation.

**αCD40-induced TLS resembled lymphoid tissues**. We further characterized TLS composition and maturity by staining for a set of well-established markers. The TLS included CD3$^+$ T cells and were dominated by B cells, of which a large proportion expressed the follicular B cell marker CD23 (Fig. 1g). We rarely observed proliferating B cells inside the TLS, but a substantial proportion of CD3$^+$ cells were Ki67$^+$, indicating T cell functionality (Fig. 1h and Supplementary Fig. 1l). A staining for the antigen binding fragment (Fab) portion of mouse IgG revealed that most B220$^+$ B cells were IgG$^+$ antibody-producing cells and that TLS contained few IgG$^+$B220$^{low/-}$ plasma cells (Supplementary Fig. 1d), suggesting that antigen selection may have occurred within these structures. The TLS contained F4/80$^+$ macrophages (Supplementary Fig. 1e), CD11c$^+$ DCs (Fig. 1i and Supplementary Fig. 1m) and CD35$^+$ or CD21$^+$ follicular dendritic cells (FDCs), which formed intimate connections with the surrounding B cells (Fig. 1j, k). Moreover, the presence of rare CD11c$^+$GFP$^+$ DCs within the TLS indicated that these cells phagocytosed GFP-positive tumor cell contents (Supplementary Fig. 1f). T regulatory cells (Tregs) were also observed in the TLS (Supplementary Fig. 1g).

TLS formed around CD31$^+$ vessels that varied in size (Fig. 1l, n and Supplementary Fig. 1n–o) and were surrounded by a distinct network of collagen IV (Fig. 1n and Supplementary Fig. 1n) and fibronectin (Fig. 1o and Supplementary Fig. 1o). The majority of B cells inside these structures stained positive for CD62L (Fig. 1m), a selectin that mediates infiltration of naïve leukocytes into lymphoid tissues[29].

The TLS varied in size, ranging from small and poorly organized clusters (Fig. 1n) to large aggregates with a follicle-like structure (Fig. 1i), where T cells were predominantly located outside the B cell zone facing the tumor tissue. TLS were present more than 2 weeks after the last administration of αCD40, indicating that continuous treatment was not required for TLS persistence (Supplementary Fig. 1h, i).

To characterize gene expression signatures associated with TLS[30], we laser capture micro-dissected TLS from αCD40-treated GL261 tumors and isolated RNA from the collected tissue (Supplementary Fig. 1j). Genes for lymphotoxin β (Ltb), C–X–C motif chemokine ligand 13 (Cxcl13), and C–C motif chemokine ligand 19 (Ccl19) were highly expressed in the TLS compared with the tumor or healthy tissue dissected from the same brain (Fig. 1p–r), while the gene for C–C motif chemokine ligand 21 (Ccl21) was expressed at a similar level (Supplementary Fig. 1k).

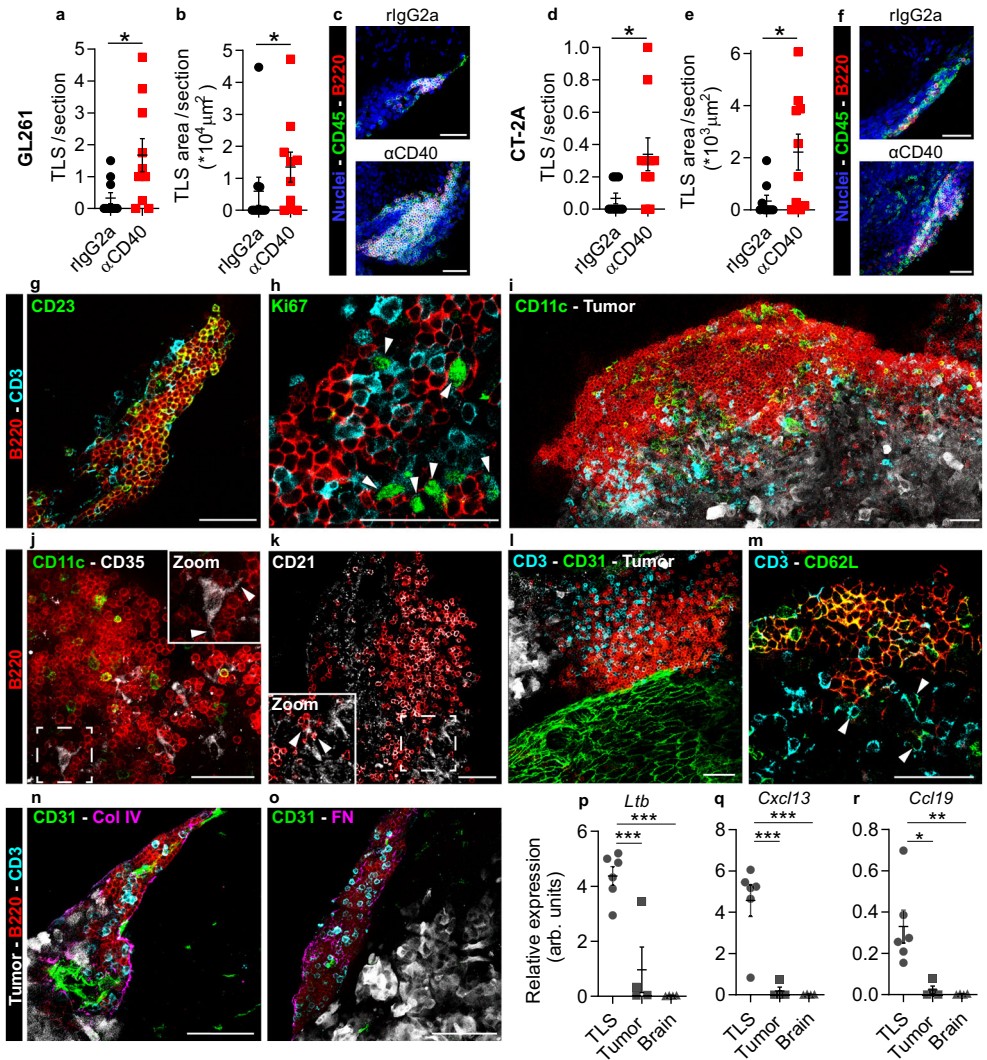

**Fig. 1 αCD40 induced formation of tertiary lymphoid structures (TLS) in the brain of GL261 and CT-2A glioma-bearing mice.** Quantification of (**a**, **d**) the number (p = 0.0166; p = 0.0106 respectively) and (**b**, **e**) the total CD45+ surface area (p = 0.0415; p = 0.0216, respectively) of dense CD45+B220+ clusters per 80μm-thick section, in the brain of GL261 and CT-2A tumor-bearing mice. In **a**, **b**, n = 10mice/group. In **d**, **e**, n(rIgG2a) = 9 mice, n(αCD40) = 10 mice. Two-tailed Mann–Whitney test. **c**, **f** Representative immunofluorescent stainings of the quantified CD45+B220+ clusters in the GL261 and CT-2A models. Scale bars: 50 μm. **g**–**o** Immunofluorescent stainings of αCD40-induced TLS in the GL261 model showing TLS composition and organization. All images are representative of four independent experiments and 8 mice, and were taken from samples collected at the survival endpoint, between day 23 and day 35 post-tumor implantation (4–16 days after the last αCD40 treatment). Arrows in **h** indicate Ki67+ T cells. Arrows in **j** indicate dendrites of a CD35+ FDC interacting with surrounding B cells. Arrows in **k** indicate a CD21+ FDC interacting with surrounding B cells. Scale bars: 50 μm. **p**–**r** Gene expression of TLS-inducing cytokines in laser capture micro-dissected CD45+B220+ clusters, compared with laser capture micro-dissected tumor tissue and normal brain tissue. arb. units = arbitrary units. n(TLS) = 6 LMD areas, n(tumor) = 4 LMD areas, n(brain) = 4 LMD areas. **p** p(TLS vs. tumor) = 0.0009, p(TLS vs. brain) = 0.0001. **q** p(TLS vs. tumor) = 0.0007, p(TLS vs. brain) = 0.0005. **r** p(TLS vs. tumor) = 0.0115, p(TLS vs. brain) = 0.0077. One-way ANOVA with Tukey's multiple comparison correction. In **a**, **b**, **d**, **e**, black circle indicates rat IgG2a (rIgG2a) and red square indicates agonistic CD40 antibodies (αCD40). In **p**–**r**, black circle indicates TLS, black square indicates Tumor and black triangle indicates Healthy brain tissue. For all graphs in this figure, *p < 0.05, **p < 0.01, ***p < 0.001. Bars: mean ± SEM. Source data are provided as a Source Data file.

Altogether, αCD40 induced the formation of tertiary lymphoid structures that contained a B cell core, T cell zones, CD11c+ DCs and CD35+/CD21+ FDCs. While B cells rarely proliferated within the TLS, a large proportion expressed the follicular B cell marker CD23 and stained positive for mouse IgG, which is indicative of antibody production, B cell follicular organization, and germinal center formation.

**B cells were required for αCD40-induced TLS formation.** To investigate the mechanism through which αCD40 induced TLS in vivo, we stained tumor sections for the rat-derived αCD40

antibody. Cells that stained positive for αCD40 were observed in both TLS and tumor area (Fig. 2b, c and Supplementary Fig. 2a). The therapeutic antibody co-localized with B220+ B cells only in the TLS of αCD40-treated mice (Fig. 2a, b), suggesting that αCD40 mainly stimulated B cells in αCD40-induced TLS. A few CD11b+ cells in the TLS also stained positive for αCD40 (Fig. 2c).

To understand whether αCD40 induced the production of TLS-associated cytokines in B cells, we isolated CD19+ cells from mouse spleen and stimulated them in vitro with αCD40. After 48 h of stimulation, B cells aggregated in clusters that became progressively larger over time (Fig. 2d). αCD40 increased the expression of *Lta* and *Tnfsf14* in B cells 48 h and 72 h after

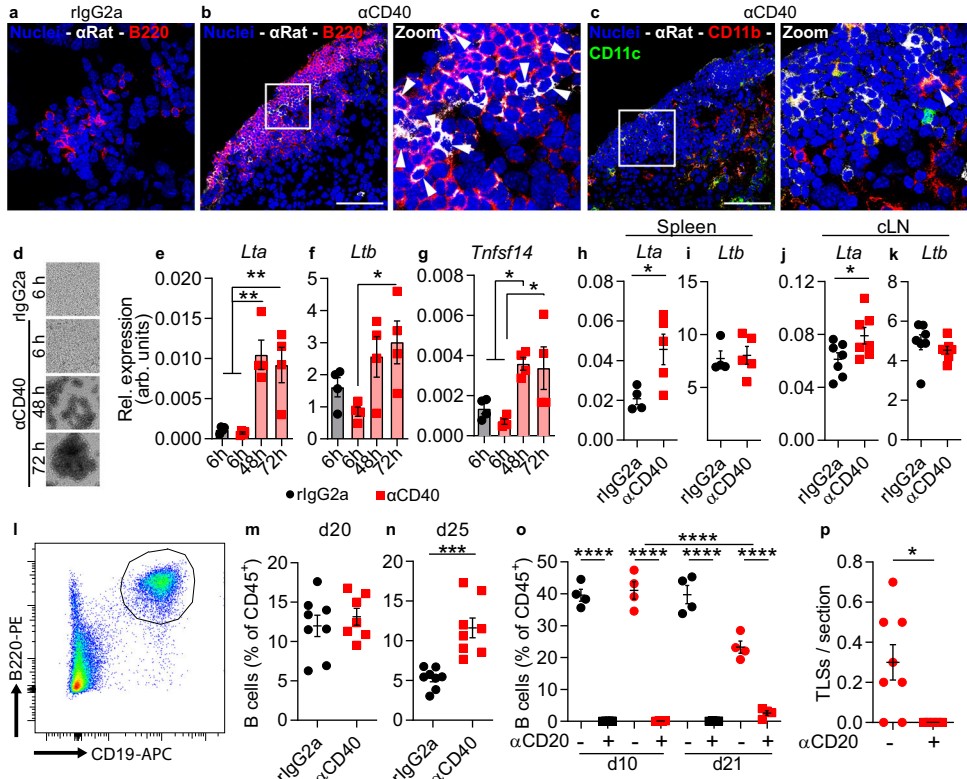

**Fig. 2 B cells expressed LTα upon αCD40 stimulation and were required for TLS formation.** All panels except (**d**–**g**) show data from GL261 tumor-bearing mice treated with rIgG2a or αCD40. **a**–**c** Immunofluorescent staining of therapeutic rat antibodies (αRat) in TLS co-stained for **a**, **b** B220 and **c** CD11b and CD11c. Images are representative of three mice. Arrows: cells positive for αRat. White square areas are magnified to the right in **b**, **c**. Scale bars: 50 μm. **d** Representative images of murine CD19$^+$ splenic B cells stimulated in vitro with rIgG2a or αCD40 antibodies at indicated time points. **e**–**g** Gene expression of *Lta*, *Ltb*, and *Tnfsf14* in B cells shown in **d**. $n = 4$ independent experiments. One-way ANOVA with Dunnett's correction for multiple comparison. In **e** $p_{(rIgG2a\ 6h\ vs.\ αCD40\ 48h)} = 0.0017$, $p_{(rIgG2a\ 6h\ vs.\ αCD40\ 72h)} = 0.005$, $p_{(αCD40\ 6h\ vs.\ αCD40\ 48h)} = 0.0012$, $p_{(αCD40\ 6h\ vs.\ αCD40\ 72h)} = 0.0035$. In **f** $p_{(αCD40\ 6h\ vs.\ αCD40\ 72h)} = 0.0223$. In **g** $p_{(rIgG2a\ 6h\ vs.\ αCD40\ 48h)} = 0.0439$, $p_{(αCD40\ 6h\ vs.\ αCD40\ 48h)} = 0.0105$, $p_{(αCD40\ 6h\ vs.\ αCD40\ 72h)} = 0.0169$. **h**–**k** Gene expression of *Lta* and *Ltb* in CD19$^+$B220$^+$ B cells sorted from **h**, **i** spleen and **j**, **k** cranial lymph nodes. **h**, **i** $n(rIgG2a) = 4$ mice, $n(αCD40) = 5$ mice. **j**, **k** $n = 7$ mice/group. **h** $p = 0.0285$, **j** $p = 0.0452$. Two-tailed *t*-test, *$p < 0.05$. **l** Representative plot of quantifications shown in **m**, **n**. **m** Quantification of CD19$^+$B220$^+$ brain-infiltrating B cells as a percentage of CD45$^+$ cells at day 20. $n(rIgG2a) = 8$ mice; $n(αCD40) = 7$ mice. **n** Quantification of CD19$^+$B220$^+$ brain-infiltrating B cells as a percentage of CD45$^+$ cells at day 25. $n = 8$ mice/group. $p = 0.0003$. **m**, **n** Two-tailed *t*-test. **o** Quantification of CD19$^+$B220$^+$ B cells as a percentage of CD45$^+$ cells in the blood of tumor-bearing mice treated with rIgG2a or αCD40 antibodies, with (+) or without (−) B cell depletion with an αCD20 antibody. $n = 4$ mice/group. All indicated p-values are $p < 0.0001$. One-way ANOVA with Tukey's correction for multiple comparison. **p** Quantification of the number of dense CD45$^+$B220$^+$ clusters per 80 μm-thick section in αCD40-treated tumor-bearing brains with (+) or without (−) B cell depletion with an αCD20 antibody. $n(αCD40) = 8$ mice, $n(αCD20 + αCD40) = 5$ mice. *$p = 0.0234$. Two-tailed *t*-test. **h**–**n** Black circle indicates rat IgG2a (rIgG2a), red square indicates agonistic CD40 antibodies (αCD40). **o**, **p** Black circle indicates rIgG2a, black square indicates rIgG2a + αCD20, red circle indicates αCD40, red square indicates αCD40 + αCD20. For all graphs in this figure, *$p < 0.05$, **$p < 0.01$, ***$p < 0.001$, ****$p < 0.0001$. Bars: mean ± SEM. arb. units = arbitrary units. Source data are provided as a Source Data file.

stimulation (Fig. 2e, g), while *Ltb* expression increased after 72 h (Fig. 2f). In line with this, B cells in the spleen and superficial cranial lymph nodes of αCD40-treated glioma-bearing mice had increased *Lta* expression (Fig. 2h, j), while *Ltb* was constitutively expressed in B cells at both locations (Fig. 2i, k). The proportion of B cells in the brain was similar across treatment groups on day 20 post-tumor implantation, while it was higher on day 25 in αCD40-treated mice compared to the rIgG2a group (Fig. 2l–n). To determine whether αCD40 stimulation of B cells was required for TLS formation, we depleted B cells 3 days before the initiation of αCD40 therapy (Fig. 2o). B cell depletion effectively inhibited the formation of TLS (Fig. 2p). In contrast, the formation of T cell aggregates characterized by a core of CD3$^+$ T cells and a network of CD11c$^+$ cells was not affected by αCD40 therapy or B cell depletion (Supplementary Fig. 2b, c). Collectively, these observations demonstrate that TLS formation was mediated by αCD40 stimulation of B cells.

## TLS were associated with increased T cell infiltration in human glioma

While αCD40 enhanced TLS formation, TLS were also present in rIgG2a-treated glioma-bearing mice (Fig. 1a–f). To determine the clinical relevance of our findings, we investigated whether similar structures were present in patients with glioma. As TLS were consistently located close to the meninges in pre-clinical glioma models, we screened patient samples that included meningeal tissue. We collected a cohort of 26 treatment-naïve patients with de-novo gliomas, which included 6 grade II gliomas, 4 grade III gliomas, and 16 grade IV glioblastomas (Supplementary Table 1).

We identified CD45$^+$CD20$^+$CD3$^+$ aggregates resembling TLS, which varied in their level of organization (Fig. 3a–n). Some clusters lacked a follicle-like organization (Fig. 3a–d), thus we defined them as "immature TLS". Some aggregates instead had a clear CD20$^+$ B cell core (Fig. 3h–k), which we defined as "organized TLS". CD35$^+$ FDCs were present in both types of TLS

(Fig. 3e, l). Occasionally, a clear CD35$^+$ FDC network was observed in organized TLS (Fig. 3l). Both TLS types included Ki67$^+$ cells (Fig. 3f, m) and formed around PNAd$^+$ HEVs (Fig. 3g, n). TLS also had rare CD23$^+$ follicular B cells (Supplementary Fig. 3a, c) and CD138$^+$ plasma cells (Supplementary Fig. 3b, d).

Immature and/or organized TLS were identified in patients with grade II/grade III glioma (3/10) and in glioblastoma patients (8/16) (Fig. 3o and Supplementary Table 1). TLS were most frequently found in close proximity to meningeal tissue, but were also observed in the white matter (close to the tumor bulk) or directly within the tumor tissue (Supplementary Fig. 3e, f). Importantly, the presence of TLS in GBM patients was associated with an increased abundance of tumor-infiltrating T cells (Fig. 3p, q). In summary, TLS were present in human glioma of various grades and were associated with an increased abundance of intratumoral T cells in GBM patients.

**αCD40 treatment resulted in impaired T cell responses**. Consistent with what we observed in human GBM, quantification of intratumoral T cells in αCD40-treated mice revealed a trend to an increased number of T cells in TLS$^+$ brains compared with TLS$^-$ brains (Fig. 4a and Supplementary Fig. 4a). However, αCD40 did not improve survival in either the GL261 or the CT-2A models (Fig. 4b, c).

CD40 stimulation is known to mediate anti-tumor immunity by inducing a CD8$^+$ T cell response via DC activation[6–8,31]. Thus, we characterized the T cell response in the tumor after αCD40 therapy using flow cytometry (Supplementary Table 4). Hierarchical stochastic neighbor embedding (HSNE) analysis indicated that CD3$^+$ T cells from GL261 and CT-2A models clustered according to the treatment regime rather than the tumor model (Fig. 4d–g). The most pronounced effects of αCD40 were observed on cytotoxic T cells (Fig. 4g; MC01, MC02), in line with a proportional increase of CD8$^+$ T cells among all brain-infiltrating immune cells (Fig. 4h). The percentage of effector CD8$^+$ T cells (CD44$^+$CD62L$^-$) was higher in αCD40-treated mice (Fig. 4i), however a greater proportion of effector cells were CD127$^-$KLRG1$^+$ (Fig. 4j), pointing to a short-lived effector phenotype[32]. A similar T cell phenotype was observed in the spleen after αCD40 treatment (Supplementary Fig. 5a–f). Markers of proliferation, exhaustion and maturation on tumor-infiltrating CD8$^+$ T cells indicated a decreased activation status upon αCD40 therapy (Fig. 4k). In line with this, αCD40-treated mice exhibited decreased percentages of CD69$^+$ and CD107a$^+$ cytotoxic T cells (Fig. 4l, m). A few model-specific responses to αCD40 were apparent, as the percentage of PD-1$^+$TIM3$^+$LAG3$^+$ cytotoxic T cells increased in the GL261 model but showed a downward trend in the CT-2A model (Fig. 4n). αCD40 did not affect the proportion of Tregs (Supplementary Fig. 4b, c). Systemically, we observed a typical cytokine response to αCD40, characterized by increased serum levels of pro-inflammatory cytokines after intravenous administration (Supplementary Fig. 5g–p).

To assess T cell functionality after αCD40 treatment, we isolated splenocytes from GL261 glioma-bearing mice and re-stimulated them in vitro with concanavalin A (ConA) for 24 h (Fig. 4o). CD8$^+$ splenocytes derived from αCD40-treated mice showed decreased proliferation, lower CD69 and a reduced percentage of CD107a$^+$ cells (Fig. 4p–r). Similar results were observed for tumor-infiltrating CD8$^+$ T cells isolated from αCD40-treated mice, which also exhibited decreased activation and proliferation (Supplementary Fig. 6a–d), and displayed impaired cytotoxicity and killing capability upon co-culture with GL261 tumor cells (Supplementary Fig. 6e–h). In summary, αCD40 induced a pro-inflammatory cytokine response consistent

with previous studies in other tumor types[33], but was associated with impaired T cell responses in the glioma microenvironment and in the spleen of glioma-bearing mice.

**αCD40 impaired the efficacy of immune checkpoint inhibitors**. We sought to understand if CPIs could rescue the αCD40-induced T cell hypofunction. GL261 glioma-bearing mice were administered four doses of αCD40 and/or anti-PD-1 blocking antibodies (αPD-1). αCD40 increased the proportion of brain-infiltrating CD8$^+$ T cells alone or in combination with αPD-1, in comparison with rIgG2a and αPD-1 monotherapy (Fig. 5a). However, the percentage of CD69$^+$Ki67$^+$ cytotoxic T cells was reduced in both αCD40-treated groups compared with αPD-1 monotherapy (Fig. 5b). Accordingly, co-administration of αCD40 and αPD-1 resulted in decreased survival compared with αPD-1 monotherapy (Fig. 5c). While αCD40 alone increased the number of TLS compared to rIgG2a, the combination regimen did not enhance TLS number compared to rIgG2a or αPD1 monotherapy (Fig. 5d). However, a substantial proportion of TLS with a larger surface area were present in co-treated mice (Supplementary Fig. 7a, b).

The circulating levels of rat-IgG were reduced in the αCD40 groups (Supplementary Fig. 7c), which may result from target-mediated clearance and/or from anti-rat-IgG responses elicited from repeated exposure to rat-IgG[31]. To exclude the possibility that drug clearance was the reason for the reduced survival in the combination group, we assessed a combination of αCD40 with a fully murine αCTLA-4 antibody and also evaluated a single dose of αCD40 followed by three doses of αPD-1 (Supplementary Fig. 7d–i). In both cases αCD40 therapy hampered the effect of the CPIs (Supplementary Fig. 7d, f). Interestingly, αCD40 was still able to induce TLS formation when combined with αCTLA-4 as compared to αCTLA-4 monotherapy.

A HSNE analysis of CD8$^+$ T cells (Fig. 5e–j) revealed that αPD-1 polarized the cytotoxic T cell response towards an active and proliferating state (Fig. 5i, j; MC02), while αCD40 resulted in impaired activation and/or proliferation (Fig. 5i, j; MC03, MC06, MC04, and MC01). Strikingly, co-administration of αCD40 and αPD-1 shifted the cytotoxic T cell response towards a low-activation, low-proliferation state (Fig. 5i, j; MC03, MC06). In summary, αCD40 induced a hypofunctional T cell state that inhibited the efficacy of CPIs.

**αCD40 activated brain-infiltrating DCs and myeloid cells**. To understand the cellular mechanisms involved in the αCD40-induced CD8$^+$ T cell hypofunction, we performed FACS analysis of tumor-infiltrating immune cells that express the CD40 molecule (Supplementary Table 4). The proportion of brain-infiltrating DCs and myeloid cells decreased after αCD40 therapy (Supplementary Fig. 8a, b). αCD40 did not enhance the production of immunosuppressive molecules such as arginase, IL-10 and PD-L1 and increased the expression of the activation marker CD86 (Supplementary Fig. 8c). αCD40 did not alter the relative amount of IL-12$^+$ DCs or myeloid cells (Supplementary Fig. 8d, e), but decreased the proportion of IL-10$^+$ DCs (Supplementary Fig. 8f, g). Altogether, αCD40 promoted an activated phenotype of brain-infiltrating DCs and myeloid cells.

**αCD40 induced CD11b-expressing B cells**. Next, we investigated the phenotype of intratumoral B cells in αCD40-treated glioma-bearing mice. αCD40 therapy increased the expression of CD86, MHC-II, and IL-12 alone or in combination with αPD-1 (Fig. 6a, b), but also increased the proportion of CD5$^+$CD1d$^+$ B cells (Supplementary Fig. 9a, b). B cells expressing CD5 and CD1d have previously been classified as regulatory B10 cells and can inhibit

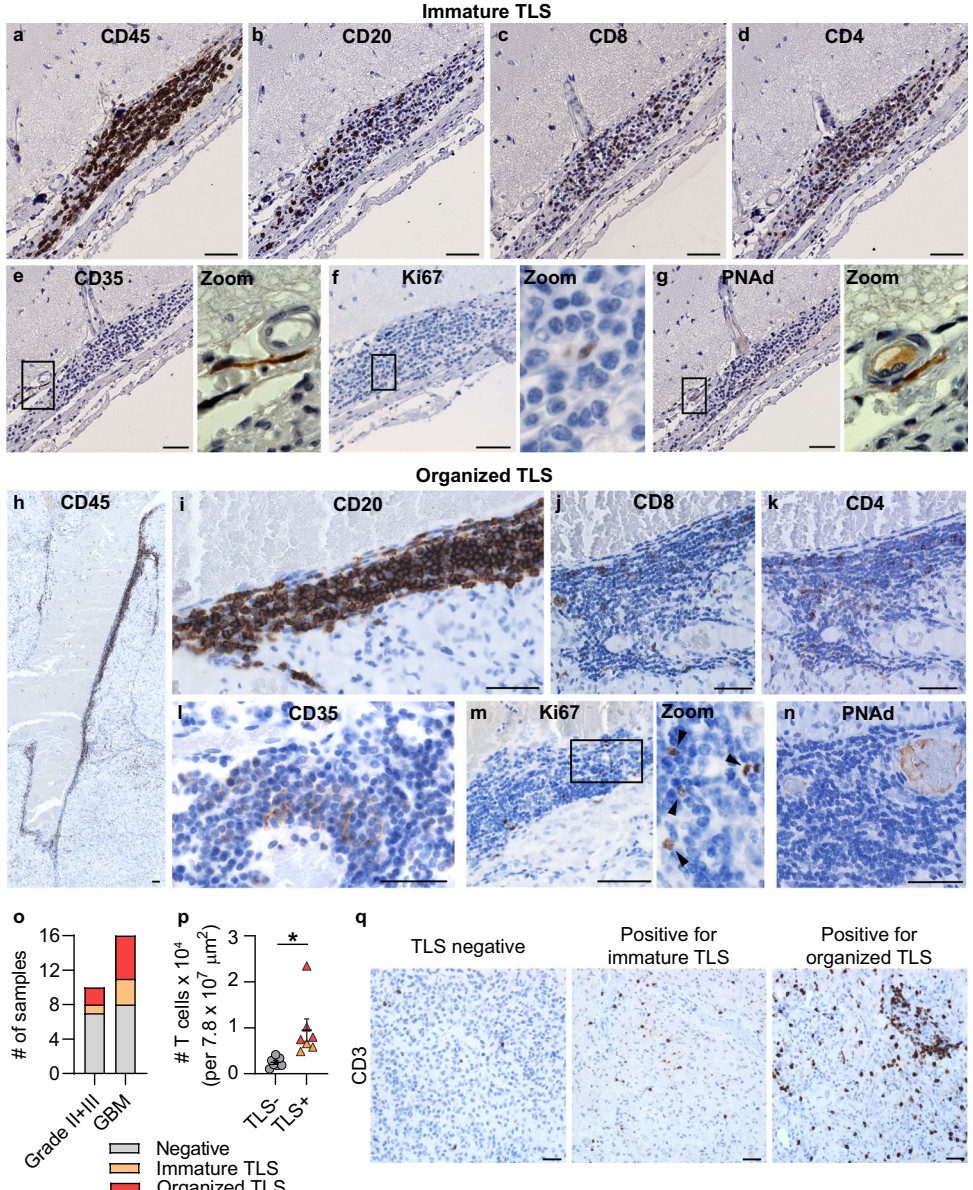

**Fig. 3 Tertiary lymphoid structures were present in the brain of glioma patients and were associated with increased T cell abundance.** Immunohistochemical stainings of human glioma sections showing the composition of (**a–g**) immature TLS characterized by a loose B cell core and (**h–n**) organized TLS characterized by a compact core of B cells. Black square areas in **e–g** and **m** are magnified to the right of each image. Scale bars: 50 μm. **a**, **b** Representative of 21 immature TLS. **h**, **i** Representative of 16 organized TLS. Stainings in **c–g** and **j–n** were performed on one representative immature TLS and one representative organized TLS. **o** Number of grade II/grade III glioma patients and glioblastoma (GBM) patients included in our cohort that stained negative for TLS (gray), positive for immature TLS (orange) or positive for organized TLS (red). $n$(Grade II + III, negative) = 7, $n$(Grade II + III, immature TLS) = 1, $n$(Grade II + III, organized TLS) = 2, $n$(GBM, negative) = 8, $n$(GBM, immature TLS) = 3, $n$(GBM, organized TLS) = 5. **p** Number of T cells infiltrating the tumor area in GBM patients negative for TLS (gray circle) versus GBM patients positive for TLS (orange triangle: immature TLS; red triangle: organized TLS). $n$ = 7 patients/group. $p$ = 0.0142. Two-tailed $t$-test. Bars: mean ± SEM. **q** Representative images of T cell infiltration in GBMs that were negative for TLS, positive for immature TLS or positive for organized TLS. Scale bars: 50 μm. For all graphs in this figure, *$p < 0.05$. Source data are provided as a Source Data file.

CD4$^+$ T cell responses via secretion of IL-10[34,35]. However, gene expression of immunosuppressive factors including *Il10*, *Tgfb1*, *Ccl22*, and *Lgals1* was not increased in B cells after αCD40 therapy (Supplementary Fig. 9c–j). In addition, production of IL-10 was increased in mice treated with αCD40 alone but not in combination with αPD-1 (Fig. 6c). Thus, it is not likely that regulatory B10 cells were the main mediators of the reduced T cell functionality.

αCD40 resulted in a striking increase of CD11b$^+$ B cells alone or in combination with αPD-1 (Fig. 6d), which have been linked to suppressed CD4$^+$ T cell responses[36,37]. A similar effect was observed in the spleen (Supplementary Fig. 10a). In the brain, CD11b$^+$ B cells were rarely observed within the TLS (Fig. 6e) but were predominantly present in the tumor area (Fig. 6f) and had lower surface levels of CD11b compared to myeloid cells (Supplementary Fig. 10b).

To understand whether CD11b upregulation was a direct effect of αCD40 stimulation of B cells or secondary to a systemic release of cytokines, we stimulated murine splenic B cells in vitro. αCD40 stimulation did not induce CD11b upregulation on B cells in vitro, while exposure to IL-10 did (Fig. 6g), consistent with previous studies[37]. Notably, IL-10 levels were systemically

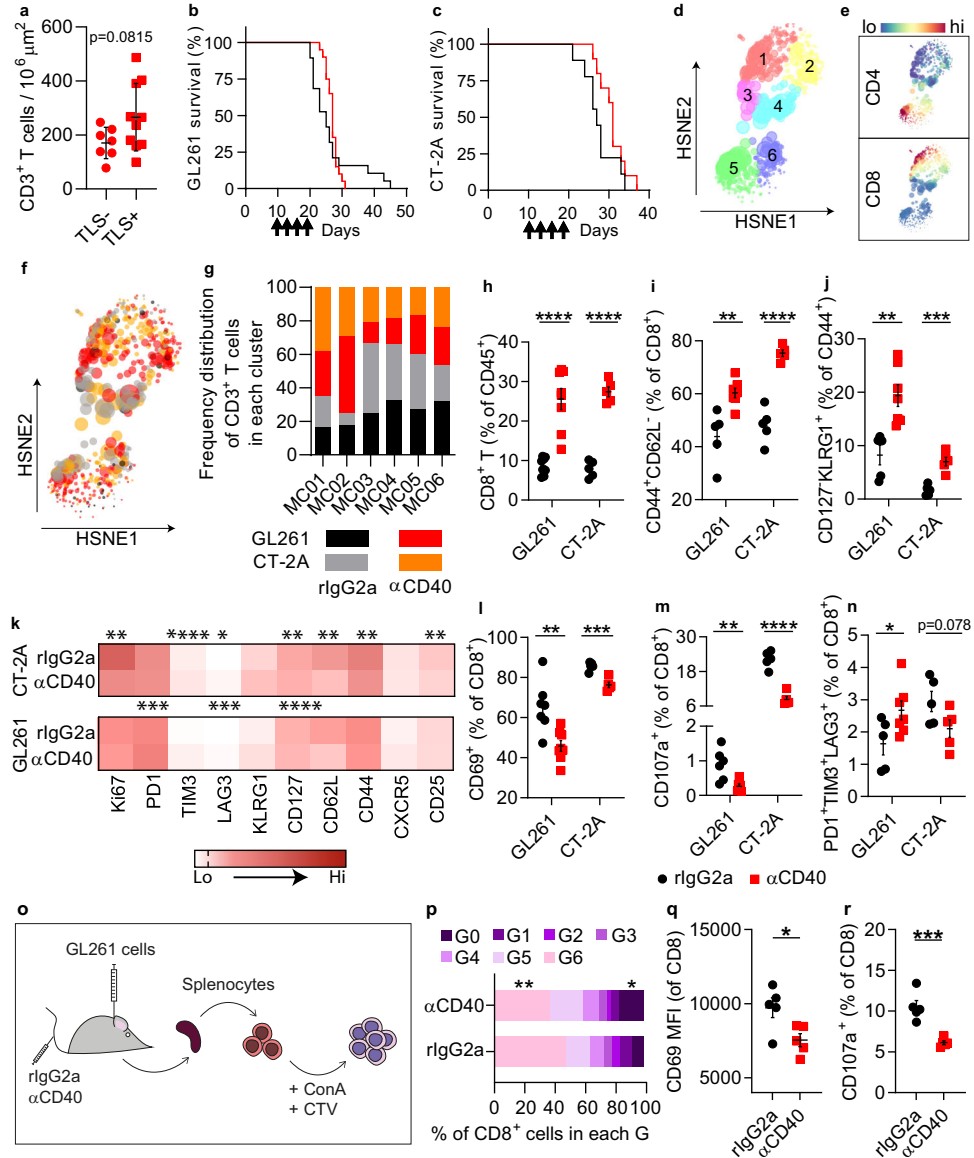

**Fig. 4 αCD40 therapy resulted in suppressed cytotoxic T cell responses in preclinical glioma models. a** Quantification of CD3[+] T cells in the tumor area of αCD40-treated GL261 glioma-bearing brains that were positive (red square) or negative (red circle) for TLS. $n$(TLS−) = 7 mice, $n$(TLS+) = 10 mice. Two-tailed $t$-test. **b**, **c** Kaplan–Meier survival curve of GL261 ($n$ = 19 mice for rIgG2a, $n$ = 20 mice for αCD40 group) and CT-2A ($n$ = 10 mice/group) tumor-bearing mice treated with αCD40 or rIgG2a on days 10, 13, 16, and 19 (as indicated by arrows). Black line: rIgG2a; Red line: αCD40. Log-rank test. **d–g** HSNE analysis of multicolor flow cytometry data showing spatial clustering of tumor-infiltrating CD3[+] cells in GL261 and CT-2A tumor-bearing mice treated with rIgG2a or αCD40. $n$(rIgG2a) = 5 mice, $n$(αCD40) = 7 mice. **d** HSNE analysis identified six meta-clusters (MC) of CD3[+] T cells, expressing different levels of CD4 and CD8 (**e**). **f** Spatial distribution of CD3[+] T cells from GL261 and CT-2A tumor-bearing brains, in rIgG2a-treated vs. αCD40-treated mice. **g** Frequency distribution of CD3[+] T cells from each model and treatment group in each MC. **h–n** Data was obtained from GL261 and CT-2A tumor-bearing mice treated with rIgG2a or αCD40. **h** CD8[+] T cells as a percentage of CD45[+] cells. $n$(GL261) = 8 mice/group, $n$(CT-2A) = 5 mice/group. $p_{(GL261)}$ < 0.0001, $p_{(CT-2A)}$ < 0.0001. **i–n** show data on CD8[+] T cells. **i** CD44[+]CD62L[−] cells, $n$(GL261-rIgG2a)&(CT-2A) = 5 mice/group; $n$(GL261-αCD40) = 7 mice. $p_{(GL261)}$ = 0.0031, $p_{(CT-2A)}$ < 0.0001. **j** CD127[−]KLRG1[+] cells, $n$(GL261-rIgG2a)&(CT-2A) = 5 mice/group; $n$(GL261 αCD40) = 7 mice. $p_{(GL261)}$ = 0.003, $p_{(CT-2A)}$ = 0.0004. **l** CD69[+] cells, $n$(GL261 rIgG2a) = 7 mice, $n$(GL261 αCD40) = 8 mice, $n$(CT-2A) = 5 mice/group. $p_{(GL261)}$ = 0.0024, $p_{(CT-2A)}$ = 0.0005. **m** CD107a[+] cells, $n$(GL261) = 6 mice/group; $n$(CT-2A) = 5 mice/group. $p_{(GL261)}$ = 0.01, $p_{(CT-2A)}$ < 0.0001. **n** PD1[+]TIM3[+]LAG3[+] cells $n$(GL261 rIgG2a) = 5 mice, $n$(GL261 αCD40) = 7 mice, $n$(CT-2A) = 5 mice/group. $p_{(GL261)}$ = 0.0462. **h–j** and **l–m** Two-tailed $t$-test. **k** Heat map showing the mean fluorescence intensity (MFI) of proliferation, exhaustion and memory markers on CD8[+] T cells. $n$(GL261 rIgG2a) = 5 mice, $n$(GL261 αCD40) = 7 mice, $n$(CT-2A) = 5 mice/group. **o** Experimental layout used to obtain data shown in panels (**p–r**). In brief, GL261 glioma-bearing mice were treated with rIgG2a or αCD40 on days 10, 13, 16, and 19 post-tumor implantation. On day 22 splenocytes were isolated, stained with cell trace violet (CTV) and re-stimulated in vitro with concanavalin A (ConA) for 24 h. **p** Percentage of CD8[+] splenocytes in different generations (G), where cells in G0 did not proliferate and cells in G6 underwent six cycles of proliferation. An example of how generations were defined is shown in Supplementary Fig. 6a. $p$(G0) = 0.0464, $p$(G6) = 0.0069. Multiple $t$-test with Sidak-Bonferroni correction. **q** Mean fluorescence intensity (MFI) of CD69 on CD8[+] splenocytes. $p$ = 0.0248. **r** CD107a[+] T cells as a percentage of CD8[+] splenocytes. $p$ = 0.0007. **q**, **r** Two-tailed $t$-test. **p–r** $n$ = 5 mice/group. For all graphs: *$p$ < 0.05, **$p$ < 0.01, ***$p$ < 0.001, ****$p$ < 0.0001. Bars: mean ± SEM. In all graphs except (**a**): black circle indicates rat IgG2a (rIgG2a), red square indicates agonistic CD40 antibodies (αCD40). Source data are provided as a Source Data file.

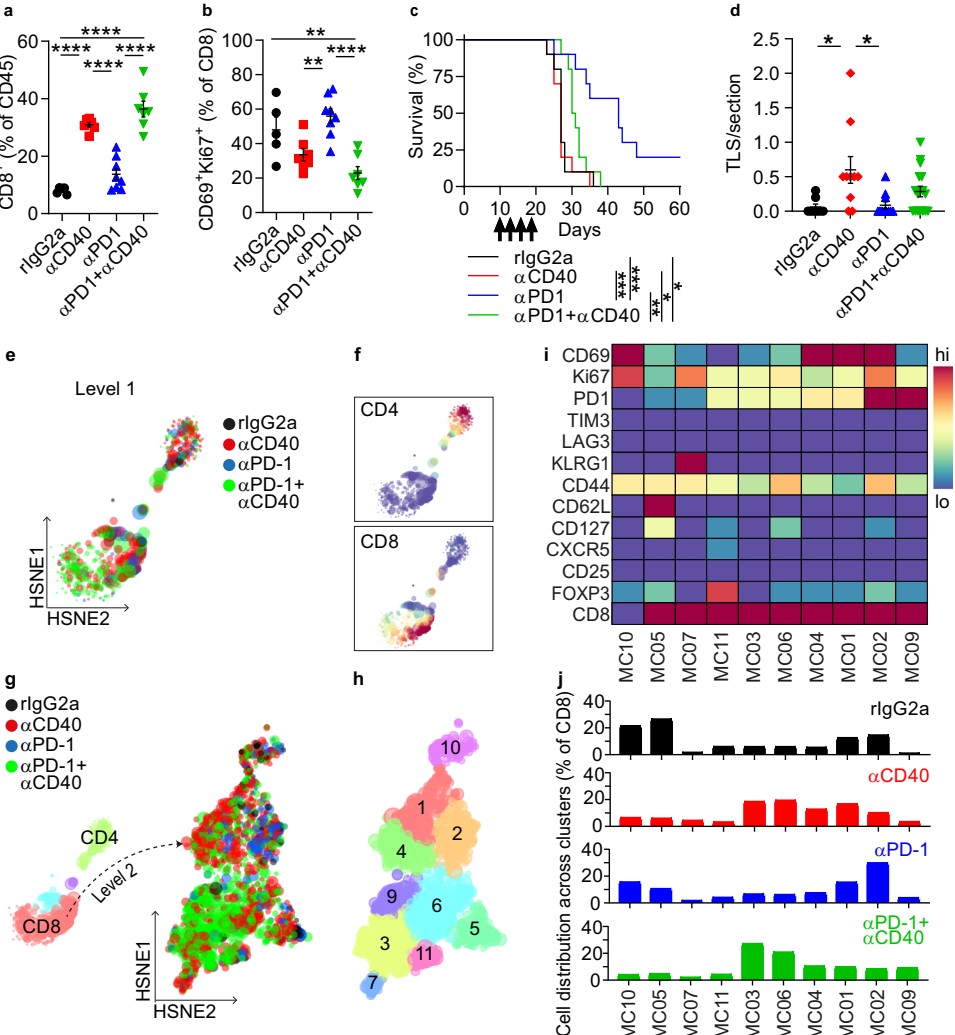

**Fig. 5 αCD40-induced T cell hypofunction impaired the efficacy of checkpoint blockade.** All panels show data from GL261 tumor-bearing mice treated with rIgG2a, αCD40, and/or αPD1. **a, b** Quantification of **a** CD8+ T cells as a percentage of CD45+ cells and **b** CD69+Ki67+ CD8 T cells in the brain in the indicated treatment groups. One-way ANOVA with Tukey's correction for multiple comparisons. **a** All indicated p-values are $p < 0.0001$. **b** $p_{(rIgG2a\ vs.\ αCD40+αPD1)} = 0.0073$, $p_{(αCD40\ vs.\ αPD1)} = 0.0064$, $p_{(αPD1\ vs.\ αCD40+αPD1)} < 0.0001$. **c** Kaplan–Meier survival curves of mice treated as indicated on days 10, 13, 16, and 19 (as shown by arrows). $n = 10$ mice/group. $p_{(rIgG2a\ vs.\ αPD1)} = 0.0008$, $p_{(rIgG2a\ vs.\ αCD40+αPD1)} = 0.0211$, $p_{(αCD40\ vs.\ αPD1)} = 0.0006$, $p_{(αPD1\ vs.\ αCD40+αPD1)} = 0.0025$, $p_{(αCD40\ vs.\ αCD40+αPD1)} = 0.037$. Log-rank test. **d** Number of dense CD45+B220+ clusters per section identified in the indicated treatment groups. $n(rIgG2a) = 8$ mice, $n(αCD40) = 10$ mice, $n(αPD1) = 11$ mice, $n(αPD1 + αCD40) = 17$ mice. $p_{(rIgG2a\ vs.\ αCD40)} = 0.0139$, $p_{(αCD40\ vs.\ αPD1)} = 0.0101$. One-way ANOVA with Tukey's correction for multiple comparisons. **e** Spatial clustering of tumor-infiltrating CD3+ cells in GL261 tumors in different treatment groups, determined by Level-1 Hierarchical Stochastic Neighbor Embedding (HSNE) analysis of flow cytometry data. **f** Expression of CD4 and CD8 across the HSNE plot shown in **e**. **g** Three meta-clusters (MC) of CD3+ T cells were identified by Level-1 HSNE analysis. Two MC were classified as mainly CD4+ or CD8+. The latter was submitted to Level-2 HSNE analysis, revealing spatial clustering of tumor-infiltrating CD8+ cells in GL261 tumors in different treatment groups. **h** MCs of T cells identified in the HSNE plot in **g**. **i** Heat map showing the expression levels of proliferation, exhaustion, and memory markers on T cells in each MC. **j** Distribution of T cells from the indicated treatment group across each MC. **a, b** and **e–j** $n(rIgG2a) = 5$ mice, $n(αCD40)\&(αPD1 + αCD40) = 7$ mice, $n(αPD1) = 8$ mice. For all graphs in this figure, *$p < 0.05$, **$p < 0.01$, ***$p < 0.001$, ****$p < 0.0001$. Bars: mean ± SEM. Black circle indicates rat IgG2a (rIgG2a), red square or diamond indicate agonistic CD40 antibodies (αCD40), blue triangle (pointing up) indicates PD1 blocking antibodies (αPD1), green triangle (pointing down) indicates αPD1 + αCD40. Source data are provided as a Source Data file.

elevated in αCD40-treated mice, with or without B cell depletion (Supplementary Fig. 10c). In conclusion, systemic delivery of αCD40 resulted in the expansion of CD11b-expressing B cells in the brain and spleen of glioma-bearing mice, which was associated with a systemic increase of IL-10.

**CD11b+ B cells inhibited CD8+ T cell responses.** Surface expression of CD11b on B cells inhibits CD4+ T cell responses, resulting in lower T cell proliferation and IFNγ production[37]. To investigate whether CD11b on B cells could suppress CD8+ T cell responses, we induced CD11b expression on B cells in vitro and

co-cultured these cells with splenocytes in the presence or absence of a CD11b-neutralizing antibody (Supplementary Fig. 11a, b). Blocking CD11b rescued activation (CD69), cytotoxicity (CD107a and IFNγ) and proliferation of CD8+ T cells stimulated with CD3/CD28 beads (Supplementary Fig. 11c–g).

Tumor-infiltrating CD11b+ B cells displayed higher levels of MHC-II compared to their CD11b− counterparts in the αCD40-treated groups (Fig. 6h), indicating an increased capability to interact with CD4+ T cells. Notably, the surface levels of CD3 were decreased on both CD4+ and CD8+ tumor-infiltrating T cells in αCD40-treated groups (Fig. 6i, j), which is in line with a CD11b-

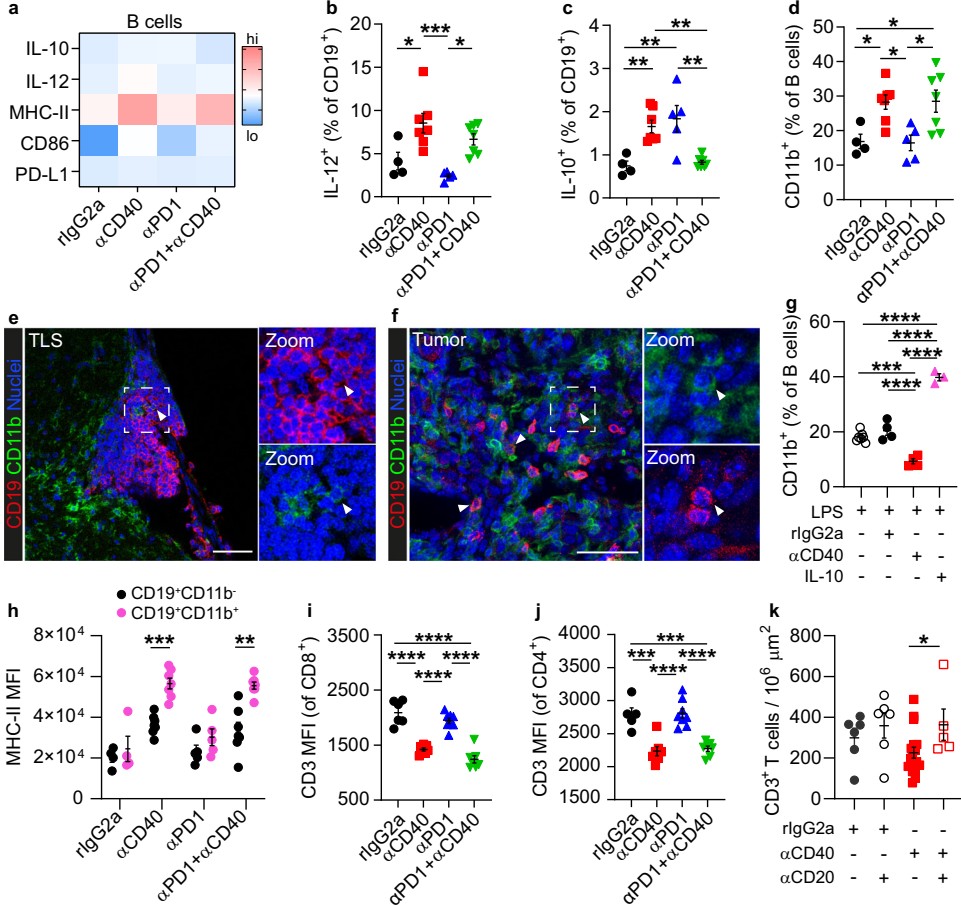

**Fig. 6 Systemic delivery of αCD40 was associated with a CD11b$^+$ regulatory phenotype of B cells.** All panels besides panel **g** show data from GL261 tumor-bearing mice. **a** Heatmap showing the expression levels of activation and immunosuppression markers on B cells in the brain, in the indicated treatment groups. **b**, **c** Quantification of **b** IL-12$^+$ and **c** IL-10$^+$ cells as a percentage of B cells in the brain, in the indicated treatment groups. **b** $p_{(rIgG2a\ vs.\ αCD40)} = 0.0175$, $p_{(αCD40\ vs.\ αPD1)} = 0.0005$, $p_{(αPD1\ vs.\ αPD1+CD40)} = 0.0140$. **c** $p_{(rIgG2a\ vs.\ αCD40)} = 0.0081$, $p_{(rIgG2a\ vs.\ αPD1)} = 0.0031$, $p_{(αCD40\ vs.\ αPD1+αCD40)} = 0.0047$, $p_{(αPD1\ vs.\ αPD1+αCD40)} = 0.0018$. **d** CD11b$^+$ cells as a percentage of B cells in the brain, in the indicated treatment groups. $p_{(rIgG2a\ vs.\ αCD40)} = 0.0453$; $p_{(rIgG2a\ vs.\ αPD1+αCD40)} = 0.0391$, $p_{(αCD40\ vs.\ αPD1)} = 0.0236$; $p_{(αPD1\ vs.\ αPD1+αCD40)} = 0.0201$. **a–d** $n(rIgG2a) = 4$ mice, $n(αCD40) = 7$ mice, $n(αPD1) = 5$ mice, $n(αPD1 + αCD40) = 7$ mice. **b–d** One-way ANOVA with Tukey's correction for multiple comparisons. **e**, **f** Immunofluorescent stainings showing CD11b$^+$ B cells **e** in the TLS and **f** in the tumor area. Scale bars: 50 μm. **g** Quantification of CD11b$^+$ cells as a percentage of wt murine splenic B cells after in vitro stimulation with the indicated agents. $n(LPS) = 7$ mice, $n(LPS + rIgG2a) = 4$ mice, $n(LPS + αCD40) = 4$ mice, $n(LPS + IL-10) = 3$ mice. One way ANOVA with Tukey's correction for multiple comparisons. $p_{(LPS\ vs.\ LPS+αCD40)} = 0.0002$, all other indicated p-values are $p < 0.0001$. **h** Mean fluorescence intensity (MFI) of MHC-II on CD11b$^+$ vs CD11b$^-$ B cells in the brain, in the indicated treatment groups. $n(rIgG2a) = 4$ mice, $n(αCD40) = 7$ mice, $n(αPD1) = 5$ mice, $n(αPD1 + αCD40) = 7$ mice. $p_{(αCD40)} = 0.0002$; $p_{(αPD1+αCD40)} = 0.0017$. Multiple t-test with Sidak-Bonferroni correction. **i, j** MFI of CD3 on **i** CD8$^+$ and **j** CD4$^+$ T cells in the indicated treatment groups. One-way ANOVA with Tukey's correction for multiple comparisons. **i** $n(rIgG2a) = 6$ mice, $n(αCD40)\&(αPD1 + αCD40) = 7$ mice, $n(αPD1) = 8$ mice. All indicated p-values are $p < 0.0001$. **j** $n(rIgG2a) = 5$ mice, $n(αCD40)\&(αPD1 + αCD40) = 7$ mice, $n(αPD1) = 8$ mice. $p_{(rIgG2a\ vs.\ αCD40)} = 0.0002$, $p_{(rIgG2a\ vs.\ αPD1+αCD40)} = 0.0004$. All indicated p-values are $p < 0.0001$. **k** Quantification of the number of CD3$^+$ T cells in the tumor area in rIgG2a- or αCD40-treated mice with or without B cell depletion. $n(rIgG2a)\&(rIgG2a + αCD20) = 6$ mice, $n(αCD40) = 17$ mice, $n(αCD40 + αCD20) = 5$ mice. $p_{(αCD40\ vs.\ αCD40-B)} = 0.0442$. One-way ANOVA with Tukey's correction for multiple comparisons. **b–d**, **i, j** Black circle indicates rat IgG2a (rIgG2a), red square indicates agonistic CD40 antibodies (αCD40), blue triangle (pointing up) indicates PD1 blocking antibodies (αPD1), green triangle (pointing down) indicates αPD1 + αCD40. **g** Empty circle indicates lipopolysaccharide (LPS), black circle indicates LPS + rat IgG2a (rIgG2a), red square indicates LPS + agonistic CD40 antibodies (αCD40), pink triangle (pointing up) indicates LPS + interleukin 10 (IL-10). **h** Black circle indicates CD19$^+$CD11b$^-$ B cells, pink circle indicated CD19$^+$CD11b$^+$. **k** Black circle indicates rIgG2a, empty circle indicates rIgG2a + αCD20, red square indicates αCD40, empty square indicates αCD40 + αCD20. For all graphs in this figure, *$p < 0.05$, **$p < 0.01$, ***$p < 0.001$, ****$p < 0.0001$. Bars: mean ± SEM. Source data are provided as a Source Data file.

mediated internalization of the TCR complex as previously reported[37]. Moreover, similarly to CD8$^+$ T cells (Fig. 5b), intratumoral CD4$^+$ T cells showed lower activation and proliferation in αCD40-treated mice (Supplementary Fig. 10d, e), and depletion of B cells prior to in vivo administration of αCD40 increased the overall abundance of intratumoral T cells (Fig. 6k). Altogether, these data suggest that CD11b$^+$ B cells underlie the suppression of T cell responses observed in αCD40-treated mice.

## Discussion

Agonistic CD40 antibodies are under clinical development for multiple solid tumors[11] and are being investigated in two Phase I clinical trials of CNS malignancies [NCT03389802; NCT04547777]. In this study, systemic delivery of αCD40 impaired T cell responses, promoted the expansion of suppressive CD11b$^+$ B cells, but also enhanced the formation of TLS in the brain.

The mechanisms involved in TLS formation during pathological conditions can vary. Besides lymphoid tissue inducer (LTi) cells, other cell types can express *Lt* to induce TLS[38,39]. B cells can act as LTi cells in the gut[38] and their transient activation via αCD40 antibodies enhanced TLS maturation in an artificial model of TLS induction[40]. Our study reveals that αCD40 stimulation of B cells promotes TLS formation in glioma-bearing mice by upregulating *Lta*. Notably, we identified TLS with varying levels of organization in the brain of patients with lower grade gliomas and GBM. Together with the fact that CD40 activation can induce *Lta* expression in human B cells[41], this strongly suggests that αCD40 could have similar effects on TLS induction in glioma patients.

Similarly to other cancer types[42], the presence of TLS was associated with increased T cell infiltration in human GBM, which could be beneficial if the T cells are activated and primed against the tumor. However, a larger cohort is needed to further elucidate the role of TLS in glioma immunity, response to checkpoint blockade and patient survival. It is also important to determine how TLS formation in the brain is affected by other cancer therapies, as corticosteroid treatment and hypofractionated radiotherapy have been associated with reduced TLS formation in other cancer types[43,44].

In αCD40-treated glioma-bearing mice, TLS also correlated with increased numbers of intratumoral T cells. However, systemic αCD40 treatment led to an accumulation of CD11b+ B cells in the brain and spleen, which was associated with impaired cytotoxic T cell responses and reduced efficacy of CPI therapy. While it is clear that innate B cells can produce regulatory cytokines[45], Liu et al. found that CD11b+ B cells can suppress CD4+ T cells through cell-to-cell interaction in a CD11b-dependent manner, inducing internalization of the T cell receptor (TCR)[37]. In line with this, intratumoral T cells in αCD40-treated mice had decreased surface levels of CD3 (suggesting internalization of the TCR complex) and B cell depletion increased the abundance of T cells in the tumor specifically in αCD40-treated mice. Moreover, blocking CD11b on B cells in vitro rescued CD8+ T cell responses, suggesting that CD11b+ B cells are capable of suppressing cytotoxic T cells.

CD11b expression was not directly induced by αCD40 stimulation of B cells. Rather, it was associated with an increase in systemic IL-10 after αCD40 treatment which was observed also when the B cells were depleted, suggesting that these cells were not the main IL-10 producers. Thus, targeting αCD40 specifically to B cells could help circumvent the upregulation of CD11b while still inducing B cell activation and expression of *Lta*. In line with this, a recent study reported that systemic administration of 4-1BBL+ B cells activated in vitro with αCD40 and IFNγ elicited anti-tumor immunity in glioma-bearing mice[46]. Contrary to what we observed in glioma models, αCD40 therapy generally enhances immune response in peripheral tumor models. Therefore, the lack of therapeutic effect in glioma is likely due to the distinct immune regulation in the brain. For instance, T cells are sequestered in the bone marrow specifically in response to intracranial tumors, leading to T cell lymphopenia which considerably affects the response to immunotherapy[47]. Moreover, B cells are particularly important for antigen presentation and T cell-mediated anti-tumor immunity in the brain[48], thus the acquisition of a suppressive CD11b+ B cell phenotype may explain the detrimental effect of αCD40 on T cell responses specifically in brain tumors.

CD11b+ B cells were rarely present within αCD40-induced TLS, therefore they are not likely to mediate immunosuppression within these structures. However, we observed T regulatory cells (Tregs) in αCD40-induced TLS. Since the presence of Tregs in TLS has been associated with suppressed anti-tumor immune

responses[49] and tumor progression[50], the role of αCD40-induced TLS in glioma has to be further investigated.

Our study demonstrates that systemic αCD40 therapy results in reduced cytotoxic T cell responses and decreases the efficacy of CPIs in preclinical glioma models. A potential limitation of this study is that the preclinical glioma models used have a high mutational burden, and are more responsive to immune checkpoint blockade than most de-novo human gliomas. Nevertheless, the induction of regulatory B cells and of TLS by αCD40 is likely to have a similar impact on the immune response in human glioma, making this relevant information for clinical trials currently investigating αCD40 therapy in patients with primary CNS tumors. Our work also reveals that TLS are present in glioma patients and that immunotherapies can modulate these structures in murine glioma models. The importance of TLS in response to CPIs is not known, and needs to be further investigated. In our study, αPD1 therapy, but not αCTLA-4, hampered the ability of αCD40 to induce TLS formation in the brain, consistent with the role of PD-1 in regulating B cell survival in germinal centers[51]. The finding that TLS in the brain can be manipulated therapeutically opens up possibilities for triggering or suppressing immune responses, which has broader implications for brain malignancies and autoimmune diseases of the central nervous system.

## Methods

**Cell lines**. The GL261[52] (gift from Dr. Geza Safrany, NRIRR, Budapest, Hungary) and CT-2A[53] (gift from Dr. T. Seyfried, Boston College, Boston, MA, USA) cell lines were transfected with lentiviruses to express GFP and luciferase[54]. GL261 cells were cultured in Dulbecco's Modified Eagle's Medium (Life Technologies, Carlsbad, CA, USA) with 10% (vol/vol) heat-inactivated fetal bovine serum (FBS) (Life Technologies). CT-2A cells were cultured in RPMI-1640 (Life Technologies) supplemented with 10% (vol/vol) heat-inactivated FBS. All cell lines were cultured at 37 °C and 5% $CO_2$ in a humidified cell incubator. The cell lines were not authenticated after purchase but routinely tested negative for mycoplasma contamination using the MycoAlert Detection Kit (Lonza, Basel, Switzerland).

**Orthotopic murine glioma models**. GL261 or CT-2A cells were orthotopically injected in mouse brains to obtain models of glioma with high mutational burden. Six to ten-week-old female C57BL/6 mice were purchased from Taconic M&B (Bomholt, Denmark) or Janvier Labs (Le Genest-Saint-Isle, France) and housed in individually ventilated Sealsafe Plus GM500 cages (IVC) (Tecniplast, Buguggiate, VA, Italy) controlled by an IVC climate system regulating humidity (~50%) and temperature (23 °C) in each cage (bedding material: aspen padding; enrichment: Bed'Rnest nesting material and paper houses). Cages and water bottles were changed once per week and the animals had free access to R36 pellets. Mice were exposed to 12 h light on/light off cycles. For injection of tumor cells, mice (at least 7 weeks of age) were anesthetized with 2.5% isoflurane and immobilized in a stereotaxic frame on a heated surface. A midline incision was made on the scalp and a hole was drilled in the skull at −1 mm anteroposterior and +1.5 mm mediolateral stereotactic coordinates from the bregma. GL261 cells ($2 \times 10^4$) or CT-2A cells ($5 \times 10^4$) were delivered in 2 μl of Dulbecco's phosphate-buffered saline (DPBS) (Thermo Fisher Scientific, Waltham, MA, USA) at a depth of 2.7 mm. The incision was closed using Vetbond tissue glue (3M, St. Paul, MN, USA) and the mice were observed until full recovery from anesthesia on a heated surface. For survival studies, mice were monitored daily and sacrificed at the appearance of tumor-induced symptoms, such as hunched posture, lethargy, persistent recumbency, and weight loss, resulting in a score of ≥0.5 according to the Uppsala University (Uppsala, Sweden) scoring system for animal welfare. All animal experiments were approved by the Uppsala County regional ethics committee (permits C1/14, C26/15, N164/15, and 5.8.18-19429/2019), and were performed according to the guidelines for animal experimentation and welfare of Uppsala University. At the survival end-point, mice were sacrificed via cervical dislocation or anesthetized for intracardiac perfusion with 10 ml of phosphate-buffered saline (PBS) (Thermo Fisher Scientific) and 10 ml of 4% (wt/vol) paraformaldehyde (PFA) (Sigma-Aldrich, St. Louis, MO, USA). At the experimental end-point (day 20–25), mice were sacrificed via cervical dislocation.

**In vivo antibody therapies**. Cages were randomly assigned to different treatment groups. Agonistic rat-anti-mouse CD40 (clone: FGK4.5, Cat# BE0016, 100 μg/dose), rat-anti-mouse PD-1 (clone: RMP1-14, Cat# BE0146, 200 μg/dose), mouse-anti-mouse CTLA-4 (clone 9D9, Cat# BE0164, 100 μg/dose), rat-anti-mouse CD20 (clone AISB12, Cat# BE0302, 200 μg/dose) antibodies (Abs) were administered intravenously in a final volume of 100 μl. αCD40 Abs were administered either in

repeated doses (on day 10, 13, 16, and 19 after tumor implantation) or in a single dose regimen (on day 9 after tumor implantation). αPD-1 antibodies were administered (a) alone, (b) in combination with αCD40 antibodies in repeated doses (on day 10, 13, 16, and 19 after tumor implantation) or (c) in a sequential treatment regimen (on day 10, 13, and 16 after tumor implantation, following the administration of a single dose of αCD40 Abs on day 9). αCTLA-4 Abs were administered alone or in combination with αCD40 Abs on day 10, 13, 16, and 19 after tumor implantation. αCD20 antibodies were administered on day 7 after tumor implantation, followed by repeated doses of αCD40 as specified above. The isotype control rat IgG2a (clone: 2A3, Cat# BE0089) was administered intravenously to the control groups in each experiment (100 μg/dose during rIgG2a vs. αCD40 experiments; 200 μg/dose during combination experiments with repeated treatment regimen; 100 μg/dose on day 9 followed by 200 μg/dose on days 10, 13, and 16 during the experiment with sequential treatment regimen). All antibodies for in vivo studies were purchased from BioXCell, Lebanon, NH, USA, and diluted in PBS.

**Isolation of immune cells from tumor-bearing mice.** Single cell suspensions of tumor-bearing brains were obtained by enzymatic dissociation of the whole brain minus the cerebellum using a gentleMACS Octo Dissociator and the Tumor Dissociation kit (Miltenyi Biotec, Bergisch Gladbach, Germany). Myelin depletion was achieved by either using Myelin Removal Beads II (Miltenyi Biotec) or by resuspending the cells in a solution of 25% BSA (in PBS) and centrifuging at 650×g for 20 min on a low brake (brake = 2) to separate the myelin ring from the cell pellet. CD45+ immune cells and CD8+ T cells were enriched using either Mouse CD45 MicroBeads (Miltenyi Biotec) or Mouse CD8 (TIL) MicroBeads (Miltenyi Biotec), respectively. Spleens were mechanically dissociated and cranial lymph nodes were digested by using 2.0 Wunsch U/ml of liberase TL (Roche, Basel, Switzerland) for 20 min at 37 °C. Lymph nodes and spleens were subsequently passed through a 70 μm strainer (Corning, Sigma-Aldrich, St. Louis, MO, USA) in PBS to obtain a single cell suspension. After isolation, cells were used for gene expression analysis, flow cytometry, FACS or ex vivo assays. For the ex vivo stimulation experiment, the isolated CD45+ cells were instead cultured with PMA (50 ng/ml), ionomycin (500 ng/ml), and Brefeldin A (1 μg/ml) for 5 h (Leukocyte Activation Cocktail, with BD GolgiPlug, BD Biosciences, San Jose, CA, USA).

**Ex vivo T cell functionality assays.** All T cell functionality assays were performed in 96-well plates in T cell medium: RPMI 1640 (Life Technologies, Carlsbad, CA, USA) added with 10% FBS, 2 mM L-glutamine, 10 mM HEPES, 20 μm β-mercaptoethanol, 1 mM sodium-pyruvate, 100 U/ml penicillin-streptomycin (all purchased from Thermo Fisher Scientific, Waltham, MA, USA) and 100 IU/ml IL-2 (Novartis, Basel, Switzerland). T cell assays were performed with cells isolated from GL261 glioma-bearing mice. To achieve ex vivo stimulation of brain-infiltrating CD8+ T cells or splenocytes, cells were isolated on day 22 post-tumor implantation (3 days after the last αCD40 treatment on day 19) and cultured in T cell medium added with 2 μg/ml of concanavalin A (Sigma-Aldrich, St. Louis, MO, USA). Splenocytes were stimulated for 24 h. CD8+ TILs were stimulated for 24 and 72 h. Before plating, cells were stained using the CellTrace™ Violet Cell Proliferation Kit (Thermo Fisher Scientific) following the manufacturer's instructions. At each time-point, cells were collected and stained to assess activation status (CD69) and proliferation status (cell trace violet) by flow cytometry.

To assess T cell functionality ex vivo, brain-infiltrating CD45+ immune cells were isolated on day 22 of post-tumor implantation (three days after the last αCD40 treatment on day 19) and were co-cultured with GL261 cells expressing luciferase at a 7:1 ratio (immune cells: tumor cells). Cells were co-cultured for 24 and 72 h in T cell medium. At each time-point, cells were collected and stained to assess IFNγ production and degranulation (CD107a) of CD8+ T cells by flow cytometry. Viability of GL261 cells at 72 h was assessed using the ONE-Glo™ Luciferase Assay System (Promega, Madison, WI, USA) following the instructions of the manufacturer.

**Isolation of splenic B cells from wt mice.** Primary mouse B cells were isolated from the spleen of 8-week-old to 12-week-old C57BL/6 mice (males or females, bred in-house). Spleens were mechanically dissociated and passed through a 70 μm cell strainer (Corning, Sigma-Aldrich, St. Louis, MO, USA) to obtain a single-cell suspension in sterile PBS. Splenic B cells were isolated by positive selection using Mouse CD19 MicroBeads (Miltenyi Biotec, Bergisch Gladbach, Germany) according to the manufacturer's instructions, and were confirmed ≥98% positive for B220 by flow cytometry.

**In vitro stimulation of splenic B cells.** To investigate gene expression after αCD40 stimulation, murine splenic B cells were plated at 2.5 × 10⁶/ml in 24-well plates in B cell medium: RMPI 1640 + 10% FBS (Life Technologies, Carlsbad, CA, USA) + penicillin-streptomycin (100 U/ml, Thermo Fisher Scientific, Waltham, MA, USA). Cells were incubated with 10 μg/ml of αCD40 (clone: FGK4.5) (BioXCell, Lebanon, NH, USA) or rat IgG2a (clone: 2A3, BioXCell) for 6, 24, 48 h, and collected for gene expression analysis.

To investigate whether CD40 stimulation or IL-10 affected CD11b expression, murine splenic B cells were plated at 2.0 × 10⁶ cells/ml in B cell medium + 5 mM

of Mg²⁺ in 96-well plates. Cells were incubated for 48 h with (a) medium alone, (b) 2 μg/ml of lipopolysaccharide (LPS, Sigma-Aldrich, St. Louis, MO, USA), (c) 2 μg/ml of LPS + 10 μg/ml of rat IgG2a (clone: 2A3, BioXCell), (d) 2 μg/ml of LPS + 10 μg/ml of αCD40 (clone: FGK4.5, BioXCell), (e) 2 μg/ml of LPS + 50 ng/ml of IL-10 (Recombinant mouse IL-10, Biolegend, San Diego, CA, USA). After 48 h, cells were collected and stained to assess CD11b surface expression by flow cytometry.

**CD11b inhibition assay.** To induce CD11b expression on B cells in vitro, murine B cells were isolated from wt spleens and incubated with LPS for 48 h as indicated above. The remaining splenocytes were cultured for 48 h in T cell medium to promote T cell expansion. After 48 h, T cells were stained using the CellTrace™ Violet Cell Proliferation Kit (Thermo Fisher Scientific, Waltham, MA, USA) and incubated with MACSiBeads αCD3/CD28 (Miltenyi Biotec, Bergisch Gladbach, Germany) following the instructions of the manufacturers. The percentage of CD11b+ B cells was assessed by flow cytometry and B cells and T cells were co-cultured at a T cell to CD11b+ B cell ratio of 1: 1. Cells were co-cultured for 72 h in the presence of an αCD11b neutralizing antibody (clone M1/70, 10 μg/ml, Biolegend, San Diego, CA, USA) or a control rIgG2b k antibody (clone RTK4530, 10 μg/ml, Biolegend). After 72 h, cells were collected and stained to assess expression levels of CD69, CD107a, IFNγ, and proliferation of CD8+ T cells by flow cytometry.

**RNA isolation, cDNA synthesis, and qPCR.** Samples from cell culture, laser capture microdissection, and FACS were collected in RLT lysis buffer (Qiagen, Hilden, Germany). RNA was isolated using the RNeasy Plus Mini or RNeasy Micro kits (Qiagen), according to the manufacturer's instructions. Reverse transcription into cDNA was performed using the SuperScript III kit (Life Technologies, Carlsbad, CA, USA). Since the number of sorted B cells was low, cDNA was pre-amplified using the SsoAdvanced™ PreAmp Supermix (Biorad, Hercules, CA, USA). qPCR was performed using 2× SYBR Green PCR Master Mix (Life Technologies) in MicroAmp® Optical 96-well Reaction Plates (Applied Biosystem, Foster City, CA, USA) with 0.25 mM sense and antisense primers per well (final reaction volume 20 μl). Primer sequences can be found in Supplementary Table 2. Plates were run in a QuantStudio3 Real-Time PCR machine (Applied Biosystem) and data was collected using QuantStudio 3 v1.4.3. Relative gene expression compared to that of *Hprt* was calculated using the ΔCT method[55].

**Flow cytometry and FACS.** Cells were stained using a live-dead dye (Supplementary Tables 4 and 5) following the instructions of the manufacturer. Unspecific Fc receptor binding in all single-cell suspensions was blocked by using anti-mouse CD16/CD32 antibody (clone 93, Biolegend, San Diego, CA, USA). Cells were stained for the markers of interest using fluorochrome-conjugated antibodies (Supplementary Tables 4 and 5). All antibodies were diluted from stock concentration according to the ratios reported in Supplementary Tables 4 and 5. For staining of FoxP3, the FOXP3 Fix/Perm Buffer Set (BioLegend) was used following the instructions of the manufacturer. For intracellular cytokine staining, the eBioscience™ Invitrogen™ Intracellular Fixation & Permeabilization Buffer Set (Thermo Fisher Scientific, Waltham, MA, USA) was used following the instructions of the manufacturer. Samples were run on FACSCanto II, LSR Fortessa (BD BioSciences, San Jose, CA, USA) (data was collected using BD FACSDiva 8.0.2) or CytoFLEX LX (Beckman Coulter, Brea, CA, USA) (data was collected using CytExpert 2.1); alternatively, the cells were sorted directly into RLT lysis buffer (Qiagen) using FACS AriaIII (BD BioSciences). Data were analyzed using FlowJo version 10.5.3 (FlowJo LLC, Ashland, OR, USA) or Cytosplore version 2.2.1[56,57]. Gating strategies used in this paper can be found in Supplementary Figs. 12 and 13.

**HSNE analysis.** HSNE analysis was performed using Cytosplore version 2.2.1[56,57]. Data obtained from a T cell multicolor FACS panel (17 colors, Supplementary Table 4) were initially analyzed by using FlowJo version 10.5.3 (FlowJo LLC, Ashland, OR, USA) to select CD45+CD3+ live cells. The data were then uploaded to Cytosplore version 2.2.1[56,57] and a Hierarchical Stochastic Neighbor Embedding (HSNE) analysis was performed on non-transformed data (number of scales = 5) to identify clusters of T cells with different phenotypes. The following active markers were selected: CD4, CD8, CD69, Ki67, PD-1, TIM3, LAG3, KLRG1, CD44, CD62L, CD127, CXCR5, CD25, and FOXP3, for level-1 analysis. Clustering was performed to identify populations of CD4+ and CD8+ T cells among all CD3+ T cells. The meta-cluster in which CD8+ were highly represented was submitted to level-2 analysis, to study the cytotoxic T cell response.

**Cytokine analysis.** Serum was collected on days 13, 19, and 25 post-tumor implantation from rIgG2a-treated and αCD40-treated mice in Microvette CB300 Capillary Blood Collection Tubes (Sarstedt, Nümbrecht, Germany). Serum samples were analyzed by using a customized U-PLEX plate (Meso Scale Discovery, Rockville, MD, USA) to measure the absolute concentration of the following cytokines and chemokines: IL-6, IL-10, IL-12p70, TNF-α, IFN-γ, and CXCL10. The V-PLEX Th17 Panel 1 Mouse kit (Meso Scale Discovery) was used to determined serum concentrations of IL-16 and IL-23. The assays were performed following the protocol provided by the manufacturer. Briefly, the U-PLEX plate was coated with

capture antibodies directed against the above-mentioned targets. For both U-PLEX and V-PLEX plates, the capture antibodies were incubated with the serum samples and sulfo-tag labeled detection antibodies were used to detect the target proteins. The plates were analyzed using a Sector™ Imager 2400 (Meso Scale Discovery) and final protein concentrations were calculated by the DISCOVERY WORKBENCH software version 4.0 (Meso Scale Discovery) using a standard curve.

**Quantification of rat IgG in serum**. Serum samples were collected on day 19 of post tumor implantation from mice treated with rIgG2a, αCD40, αPD-1, and αCD40 + αPD-1 antibodies, using Microvette CB300 Capillary Blood Collection Tubes (Sarstedt, Nümbrecht, Germany). The IgG (Total) Rat Uncoated ELISA Kit with Plates kit (Thermo Fisher Scientific, Waltham, MA, USA) was used to quantify the amount of rat antibodies in serum samples, following the instructions provided by the manufacturer.

**Immunofluorescent staining of mouse samples**. After intracardiac perfusion, brains were collected, fixed overnight in 4% (wt/vol) paraformaldehyde (PFA) (Sigma-Aldrich, St. Louis, MO, USA) and cryoprotected in 30% (wt/vol) sucrose overnight. Vibratome sections (80 μm-thick) were prepared from PFA-fixed brains. Vibratome slides were permeabilized in PBS containing 0.1% Triton-X100, followed by blocking in PBS containing 3% (wt/vol) bovine serum albumin and 3% FBS (vol/vol). After cervical dislocation, brains were collected and snap-frozen in isopentane. Cryosections (7 μm-thick) were prepared from snap-frozen brain tissue and fixed in ice-cold acetone (Sigma-Aldrich) for 10 min. Frozen slides were blocked in 3% (wt/vol) bovine serum albumin in PBS for 1 h. Sections were stained using primary antibodies directed against the proteins of interest (Supplementary Table 3). Nuclear staining was performed with Hoechst 33342 (Sigma-Aldrich). The slides were mounted using Fluoromount-G (Southern Biotechnology, Birmingham, AL, USA). Images were acquired using an inverted fluorescence confocal microscope (Leica SP8, Leica Microsystems, Wetzlar, Germany) or with a Zeiss Axioimager microscope (Zeiss, Oberkochen, Germany). Leica Application Suite X 3.6.0.20104, ImageJ version 1.51[58] (NIH, Bethesda, MD, USA) or CellProfiler version 3.1.9[59] (Broad Institute, Cambridge, MA, USA) were used for image analysis and quantifications. All immunofluorescence images showing TLS composition are representative of three or more structures analyzed. Quantification of T cell numbers in the tumor area was performed on vibratome sections from brains of glioma-bearing mice. A Leica SP28 confocal microscope (Leica Microsystems) was used to take six random images for each tumor sample at ×25 magnification, of which three were taken in the tumor core and three included the tumor rim. T cells were counted using CellProfiler version 3.1.9.

**Laser capture microdissection**. Cryosections (10 μm-thick) were placed on RNAse-free POL membrane frame slides (Leica Microsystems, Wetzlar, Germany) and fixed for 2 min in ice-cold acetone (Sigma-Aldrich, St. Louis, MO, USA). The sections were stained directly with conjugated antibodies against B220 and CD45 (Supplementary Table 3), together with DAPI nuclear stain (Thermo Fisher Scientific, Waltham, MA, USA) for 1 min. Slides were rinsed with diethyl dicarbonate-treated PBS and dried before laser capture microdissection. TLS, tumor tissue and healthy brain tissue were microdissected using a Leica LMD6000 B microscope (Leica Microsystems) and collected in the cap of an RNAse-free 0.5 ml tube (Thermo Fisher Scientific, Waltham, MA, USA) in RLT lysis buffer (Qiagen, Hilden, Germany).

**Tumor material from glioma patients**. A cohort of 26 human glioma samples was assembled, which included cases of grade II glioma, grade III glioma, and grade IV glioblastoma as indicated in Supplementary Table 1. All samples were collected during surgery. All relevant ethical regulations for work with human participants have been followed and informed consent has been obtained. The study was authorized by the regional Ethics Committee of Uppsala, Sweden (DNR 2010/291).

Eleven samples were *en bloc* resected tumors from patients with suspected low-grade glioma or glioblastoma. In brief, preoperative T2-FLAIR MRI sequences were recorded to delineate the tumor border. Subsequently, microsurgical *en bloc* resections were performed of the entire tumor volume, as recorded by T2-FLAIR MRI sequences, including a margin of 1–2 cm outside the radiological border, and extending into the normal-appearing brain tissue[60]. The institutional review board at the Uppsala University Hospital (Uppsala, Sweden) approved the study (DNR2010/05).

Thirteen samples of supratentorial glioblastomas were identified in the database of Department of Surgical Pathology, Uppsala University Hospital. Samples from stereotactic biopsy specimens, small biopsies or samples that did not present with viable meningeal coverings were excluded, leaving 13 subjects within the cohort. The study was authorized by the regional Ethics Committee of Uppsala, Sweden (DNR2015/089).

Two samples were collected during supramarginal glioblastoma resection where biobanking of tissue was approved by the Ethics Committee of Western Sweden (EPN/DNR: 559-12).

Assessment of histological samples and WHO classification was performed by neuropathologists S.L. and T.O.B.

**Immunohistochemical staining of human samples**. For all samples, tissue was fixed in formalin for at least 2 days. For large biopsies, representative 3 mm-thick blocks were embedded in paraffin. For *en bloc* resected tissue, samples were cut into 6–8 mm-thick blocks covering the entire tumor volume. This was followed by an additional formalin fixation for 24 h followed by paraffin-embedding. Sequential paraffin-embedded 4 μm-thick sections were stained using a Dako Autostainer Plus (DakoCytomation, Glostrup, Denmark) and Dako EnVision FLEX detection system (DakoCytomation). Slides were pretreated in target retrieval solution (S2367, Dako) or 1× citrate (S2031, Dako) in a pressure cooker and subsequently incubated with primary antibodies against IDH1-R132H (Dianova, #DIA-H09, RRID: AB_2335716, 1:100 dilution), CD45 (#M0701, Dako, RRID:AB_2314143, 1:500 dilution), CD20 (#IR604, Dako, not diluted), CD8 (#IR623, Dako, not diluted), CD4 (#M7310, Dako, RRID:AB_2728838, 1:40 dilution), CD35 (#M0846, Dako, RRID:AB_2085149, 1:25 dilution), CD138 (#M7228, Dako, RRID:AB_2254116, 1:100 dilution), CD23 (#M7312, Dako, 1:50 dilution), PNAd (#120803, BioLegend, San Diego, CA, USA, RRID:AB_493556, 1:100 dilution), and Ki67 (#IS626, Dako, not diluted) for 30 min at room temperature. Signal was developed using the MACH3 Mouse HRP Polymer detection kit (M3M530L, Biocare Medical, Pacheco, CA, USA) and slides were counterstained with hematoxylin. Stained samples were assessed for TLS presence by L.v.H., A.V., M.R., and A.D in collaboration with a neuropathologist (S.L.). Images were collected using a Zeiss Axioimager microscope or with an Axio Scan.Z1 (Zeiss, Oberkochen, Germany), and image analysis was performed using ImageScope v12.3.3.5048 or QuPath version 0.1.2[61] and T cells in the tumor were counted using the "positive cell detection" algorithm.

**Statistical analysis**. All statistical analysis was performed using GraphPad Prism 6.0 (GraphPad, La Jolla, CA, USA). Kaplan–Meier curves were analyzed by using the log-rank test. For all other analyses, two-tailed *t*-test, two-tailed Mann-Whitney test or one-way ANOVA with Tukey's correction for multiple comparison were used to determine statistically significant differences between two or more groups, respectively. For the cytokine analysis at multiple timepoints, multiple *t*-test analysis with Sidak-Bonferroni correction was performed. The significance threshold was set at $p \leq 0.05$. The *p*-value is indicated as *$p \leq 0.05$, **$p \leq 0.01$, ***$p \leq 0.001$, and ****$p \leq 0.0001$.

**Reporting summary**. Further information on research design is available in the Nature Research Reporting Summary linked to this article.

## Data availability
Source data are provided with this paper. Additional raw image data are available from the authors upon request. The remaining data are available within the Article or Supplementary Information.

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

## Acknowledgements

We thank Patricia Fuenzalida and Minttu-Maria Martikainen for excellent technical assistance, and the SciLifeLab BioVis Core Facility (Uppsala University) for assistance with microscopy and flow cytometry. This work was supported by grants from the Swedish Cancer Society (CAN 2017/502, 20 1008 PjF, 20 1010 UsF); the Swedish Childhood Cancer Society (PR2015-0133, NCP2015-0075, and PR2018-0148); the Swedish Research Council (Dnr 2016-02495, Dnr 2020-02563); Knut and Alice Wallenberg foundation (Dnr KAW 2019.0088) and Emil and Wera Cornells Stiftelse. S.M.M. was supported by a young investigator grant by the Swedish Society for Medical Research (SSMF) (S15-0065). M.R. was supported by two postdoctoral grants from Barncancerfonden (TJ 2017-0004, TJ 2019-0014). A.D. was supported by a Senior Investigator Award from the Swedish Cancer Society (CAN 2015/1216).

## Author contributions

L.v.H., A.V., M.R., and A.D. conceived the project and designed the experiments. L.v.H., A.V., M.R., K.V., T.v.d.W., M.G., H.H., and I.P. performed the experiments. L.v.H.,

A.V., and M.R. analyzed the data and designed the figures. J.L. provided support for the laser capture microdissection experiments. A.S., S.L., M.Z., A.S.J., and T.O.B. provided the human glioma material. S.L. and T.O.B. assessed the clinical samples. M.E., M.H.U., M.C.I.K., and S.M.M. provided advice regarding experimental procedures and data interpretation. A.D. supervised the project. L.v.H., A.V., M.R., and A.D. interpreted the data and wrote the manuscript. All authors read and approved the final manuscript.

## Funding

## Competing interests
S.M.M. is the founder and shareholder of Immuneed AB and Vivologica AB and is the Chief Development Officer and shareholder of Ultimovacs ASA/AB. None of the mentioned companies have taken part in the study nor do they have a financial gain of the specific subject matter described herein. The other authors declare no competing interests.
