## [Peer Review File · Nature Communications]

Agonistic CD40 therapy induces tertiary lymphoid structures but impairs responses to checkpoint blockade in gliomaEditorial Note: This manuscript has been previously reviewed at another journal that is not operating a transparent peer review scheme. This document only contains reviewer comments and rebuttal letters for versions considered at Nature Communications.

REVIEWER COMMENTS

Reviewer #1 (Remarks to the Author):

The authors have answered all my questions

Reviewer #2 (Remarks to the Author):

The authors have done a very good job responding to the concerns and comments of the initial review of the manuscript. They were able to significantly increase the number of human samples including a large number of glioblastoma to look for the TLS structure and also provide preliminary suggestions that these were associated with an increased number of immune cells in the brain parenchyma or tumor. They also report that these intratumoral T cells may be functionally impaired. However, the discussion (line 305) does state "Although the role of TLS in glioma requires evaluation in a larger cohort, this suggests an association between the presence of TLS and enhanced immune responses in GBM patients". This statement would seem to be contrary to the prior findings of functional impairment and should be clarified.

Importantly, they have also done additional studies to determine the mechanism by which the CD40 agonist may attenuate the immune response in patients with glial tumors. These greatly add to the overall impact of the work.

Finally, the discussion would be enhanced if they mention potential confounders in patients that may alter the formation of TLS, such as radiation that will often include much of the meninges proximal to the tumor and the use of corticosteroids which have been shown to dramatically attenuate immune response, particularly immune checkpoint inhibitors.

Reviewer #3 (Remarks to the Author):

The revised version of the manuscript improves the description of the TLSs, and bring new data regarding these structures

The authors conclude for the presence of "not fully" mature TLS in the brains of anti-CD40 injected mice, this argument being based on detection of few Ki67+ B cells.

This is somehow contradictory with two findings :

a-the presence of FDC CD21+CD35+ structures in TLS may suggest that Ag selection occurs after B cell proliferation.

b-the human GBM situation where dense CD138+ cells are present in the TLS.

To respond more precisely to this question of maturation the authors should search for the presence of plasma cells in the TLS in the brains of the anti-CD40 treated animals.

Moreover it remains unclear how many TLS are present in tumors from control injected animals (saline) compared to rIgG2a injected ones in data presented in Fig 1 and in related supplementary figures. Could the authors be more precise ?

To be revised :

-Line 103 the presence of rare CD11c+GFP+ DCs within the TLS "indicated uptake of tumor-associated antigens" : functionality has not been shown, the second part of the sentence should be removed

-Line 104 "to a varying extent" ? this is not precise enough

-Line 153 : Small TLS were occasionally observed in untreated glioma-bearing mice (Fig. 1a-f) I do not see data from this experimental condition ; what does mean "occasionally" ? what does mean "small" ? this is not precise enough

-Line 126 : The sizes and degree of organization of these structures were variable, but were generally increased by α CD40 therapy : this could be more precise, at least size can be quantified (as illustrated in suppl Fig 7 d and e) and degree of organization qualified.

-Fig 5d : authors analyze the effect of co-administration of anti-CD40 and anti-PD1 on TLS induction : They claim that anti-PD1 impairs α CD40-mediated TLS induction : However results in Fig 5d on TLS numbers do not show statistical significance. This should be rephrased. Similar exp

in suppl Fig 7 shows slight effect on TLS numbers ; figure 7a seems to show smaller size for TLS in combination experiment.

-Suppl Fig 7 brings an interesting comparison of the effect of anti-PD1/antiCD40 and anti-CTLA4/antiCD40 on TLS induction : it is clear that there is no inhibition in the second combination in contrast to the 1st one despite the induction of resistance to anti-ICI by anti-CD40 in both combinations. Fig 7b misses the p values. Both observations may be discussed
It would have been interested to compare composition of TLS in B cells and T cells in both situations.

The new set of data bring a potential mechanism of action of anti-CD40 in glioma bearing mice, suggesting that the detrimental effect of anti-CD40 on response to anti-PD1 is systemic. With this new set of data, this story is somehow disconnected from the story on TLS induction by anti-CD40. Nevertheless, both stories are of interest.

Abstract should be re-written.

Reviewer #4 (Remarks to the Author):

The authors have made significant improvements to their manuscript. From the perspective of this Reviewer, no additional experiments are required. However, several elements of the manuscript should be clarified and discussed in further detail as suggested below:

1. Supplementary Table 1. The authors should clarify whether the human tissue samples were from treatment-naïve patients (i.e. de novo) or were obtained after recurrence and thus, following prior adjuvant therapy. This should also be clarified in the main text on page 8.
2. Extended data Fig 5. The authors should clarify in the legend when anti-CD40 was delivered.
3. Figure 5d (Line 219). The conclusion that anti-PD1 co-administration impairs anti-CD40-mediated TLS induction is not reflected in Figure 5d as indicated in the text. If by chance there actually is a statistical difference, it is not apparent that this is biologically significant (i.e. 0.5 vs 0.3 TLS/section). The increase in TLS with anti-CD40 also seems largely driven by two mice in the anti-CD40 group. The authors should re-consider their conclusion and revise accordingly.
4. Line 234. The conclusion that the T cells are dysfunctional is a bit misleading. "Reduced functional capacity" may be more appropriate as the T cells are still capable of significant proliferation, activation, and degranulation on ex vivo stim, albeit at modestly reduced levels.
5. Discussion. The authors should comment in the discussion regarding their unexpected findings that anti-CD40 impairs T cell responses in their glioma model. This is important since several other reports have now clearly demonstrated additive or even synergistic activity with combining CD40 agonists with immune checkpoint blockade. Perhaps the authors can further discuss their thoughts on whether this finding might reflect the biological impact of tumors growing in the brain on immune biology in the host and responses to a CD40 agonist. Specifically, prior work has shown that brain tumors can induce sequestration of naïve t cells in the bone marrow of mice and patients which might influence the therapeutic activity of immunotherapy (PMID 30104766).
6. Discussion. The authors should discuss key limitations to their study. For instance, in addition to the revision where they now indicate in the methods that the glioma cell lines used were of high mutational burden, they should also discuss this in the main text as a potential limitation. Specifically, the cell lines used are of high mutational burden which is not typical of de novo gliomas. In addition, their models are highly responsive to immune checkpoint blockade which is also not characteristic of brain cancers.

Point-by-point answer to reviewer's comments

Reviewer #1 (Remarks to the Author):

The authors have answered all my questions

Reply: We thank the reviewer for the previous comments and suggestions that helped us improve our paper!

Reviewer #2 (Remarks to the Author):

The authors have done a very good job responding to the concerns and comments of the initial review of the manuscript. They were able to significantly increase the number of human samples including a large number of glioblastoma to look for the TLS structure and also provide preliminary suggestions that these were associated with an increased number of immune cells in the brain parenchyma or tumor. They also report that these intratumoral T cells may be functionally impaired.

However, the discussion (line 305) does state "Although the role of TLS in glioma requires evaluation in a larger cohort, this suggests an association between the presence of TLS and enhanced immune responses in GBM patients". This statement would seem to be contrary to the prior findings of functional impairment and should be clarified.

Reply: We agree with the referee that increased T cell infiltration is not always associated with enhanced immune responses. Indeed, as pointed out by the referee, T-cell functionality was reduced in α CD40-treated mice, associated with an induction of suppressive CD11b⁺ B cells. In human glioblastoma, we have not assessed T cell functionality since we did not have access to fresh tissue and only immunostainings could be performed. We have clarified the statement in lines 312-322 to clearly discriminate between observations made in α CD40-treated mice and human glioblastoma.

Importantly, they have also done additional studies to determine the mechanism by which the CD40 agonist may attenuate the immune response in patients with glial tumors. These greatly add to the overall impact of the work.

Finally, the discussion would be enhanced if they mention potential confounders in patients that may alter the formation of TLS, such as radiation that will often include much of the meninges proximal to the tumor and the use of corticosteroids which have been shown to dramatically attenuate immune response, particularly immune checkpoint inhibitors.

Reply: We thank the reviewer for suggesting this important addition to the discussion. The patient cohort included in this study had not received treatment prior to surgery. However, corticosteroid treatment during chemotherapy have previously been associated with reduced TLS formation in lung squamous cell carcinoma (PMID 29279354) and hypo-

fractionated radiotherapy has been shown to transiently deplete TLS in a genetic mouse model of lung adenocarcinoma (PMID 30038899). The potential impact of radiation and corticosteroids on TLS formation in glioblastoma should clearly be considered, and this has been added to the discussion in in lines 315-318.

Thank you for the valuable comments and for carefully reviewing our work, we very much appreciate the time that you have put in to help us improve this manuscript!

Reviewer #3 (Remarks to the Author):

The revised version of the manuscript improves the description of the TLSs, and bring new data regarding these structures

The authors conclude for the presence of “not fully” mature TLS in the brains of anti-CD40 injected mice, this argument being based on detection of few Ki67+ B cells. This is somehow contradictory with two findings :

- a. the presence of FDC CD21+CD35+ structures in TLS may suggest that Ag selection occurs after B cell proliferation.
- b. the human GBM situation where dense CD138+ cells are present in the TLS.

To respond more precisely to this question of maturation the authors should search for the presence of plasma cells in the TLS in the brains of the anti-CD40 treated animals.

Reply:

- a. It is difficult to judge if the TLS are fully mature, and we have so far relied on the fact that very few B cells are Ki67+, suggesting a limited proliferation of B cells within these structures. However, it is true that FDCs (which are known to be involved in germinal center organization) are present within the TLS, and their expression of CD21 and CD35 indicates that antigens may be taken up and retained. This may suggest that affinity-driven Ag selection occurs. We attempted to stain for CD138 to locate plasma cells in α CD40-treated animals, but unfortunately it is notoriously difficult to stain CD138 on mouse tissue. We did not succeed to stain for CD138 in mouse tumor tissues or lymph nodes (used as positive controls). Instead, we stained for B220 in combination with an antibody detecting the F(ab')₂/Fab portion of mouse IgG to identify antibody-producing cells, and searched for B220^{low/-}IgG⁺ cells to identify plasma cells (Chapter 14 - B cell memory and plasma cell development, Molecular Biology of B Cells, Tasuku Honjo, Michael Reth, Andreas Radbruch, Frederick Alt, eds., 2015, <https://doi.org/10.1016/C2011-0-08288-1>). We optimized and verified the specificity of the staining in mouse lymph node, where we could clearly identify B220^{low/-}IgG⁺ plasma cells in the expected locations (Figure R1). Importantly, the anti-Fab antibody did not stain resting B cells within the lymph node, but was specific for plasma cells. When staining brains from α CD40-treated glioma-bearing mice, we observed an abundance of B220⁺/IgG⁺ B cells in the TLS as well as rare B220^{low/-}IgG⁺ cells (new Supplementary Fig. 1d). This suggests that B cells may mature to plasma cells within the TLS, and supports the notion that α CD40 can

induce mature TLS in glioma-bearing mice. The data and its implications are discussed in lines 103-106 and 130-133.

- b. In human gliomas, CD138⁺ plasma cells are rarely found within the TLS. Glioma cells can express CD138 (PMID: 22714920), and most of the CD138 staining seen in Supplementary Fig. 3d is actually in the tumor tissue. We are thankful that the referee made us aware that the data could be wrongly interpreted as an abundance of CD138 plasma cells within TLS. To avoid confusion, we have added a zoom-in panel that shows CD138 staining within the TLS and in the tumor tissue (New Supplementary Fig. 3d).

Figure R1. Lymph node from naïve mouse stained for B220 (red), IgG (green) and nuclei (blue). The zoom panel shows the presence of B220^{low}-IgG⁺ cells in the expected plasma cell locations.

Moreover it remains unclear how many TLS are present in tumors from control injected animals (saline) compared to rIgG2a injected ones in data presented in Fig 1 and in related supplementary figures. Could the authors be more precise ?

Reply: We use rIgG2a-treated mice as the isotype control group throughout this manuscript. Thus, when we have mentioned “control mice” we refer to the rIgG2a-treated group. To avoid misunderstanding, we have clarified this point in lines 80-83 and have carefully gone through the manuscript to make sure that this group is always referred to as “rIgG2a” group. Importantly, TLS are detected in untreated glioma-bearing mice, and the frequency is similar to that observed in rIgG2a-treated glioma-bearing mice (Figure R2).

Figure R2. Quantification of the number of TLS per section in GL261-glioma bearing mice that either received no intravenous injection (untreated, n=4) or received rIgG2a antibodies (rIgG2a, n=10). Bars: mean ± SEM. Statistics: t-test.

To be revised:

- Line 103 the presence of rare CD11c+GFP+ DCs within the TLS "indicated uptake of tumor-associated antigens" : functionality has not been shown, the second part of the sentence should be removed

Reply: We agree that we have not made any functional assessment of antigen uptake and have rephrased as follows "indicated that these cells phagocytosed GFP-positive tumor cell contents" (lines 109-110).

- Line 104 "to a varying extent" ? this is not precise enough.

Answer: Indeed, in this characterization of TLS composition we have only assessed if specific cell types are present within the TLS or not. Therefore we have rephrased as follows: "T regulatory cells (Tregs) were also observed in the TLS" (lines 110-111).

- Line 153 :Small TLS were occasionally observed in untreated glioma-bearing mice (Fig. 1a-f) I do not see data from this experimental condition ; what does mean "occasionally" ? what does mean "small" ? this is not precise enough

Reply: In the above-mentioned sentence, we are referring to the fact that TLS were not only observed in α CD40-treated animals, but were also found in the rIgG2a-isotype control treated mice. This suggests that TLS may be also be present in human samples collected from treatment-naïve glioma patients. To simplify, we have rephrased as follows: "While α CD40 enhanced TLS formation, TLS were also present in rIgG2a-treated glioma-bearing mice" (line 158-159). Data regarding the number of TLS per section, total TLS area per section and area of individual TLS are presented in Fig. 1(a-f) and Supplementary Fig. 7b.

- Line 126 : The sizes and degree of organization of these structures were variable, but were generally increased by α CD40 therapy : this could be more precise, at least size can be quantified (as illustrated in suppl Fig 7 d and e) and degree of organization qualified.

Reply: The purpose of this sentence was to summarize all the data in Figure 1 and it refers to the "number of TLS per section" and "TLS area per section" graphs shown in Figure 1(a-f), which was described in the first paragraph (line 90-92). To clarify that we are not referring to any new data which is not shown in the manuscript, the sentence has been removed from the end of the second paragraph.

-Fig 5d : authors analyze the effect of co-administration of anti-CD40 and anti-PD1 on TLS induction : They claim that anti-PD1 impairs α CD40-mediated TLS induction : However results in Fig 5d on TLS numbers do not show statistical significance. This should be rephrased. Similar exp in suppl Fig 7 shows slight effect on TLS numbers ;

Reply: It is true that we do not see a significant difference in the number of TLS/section between the α CD40 group and the combination group. However, there is also no significant increase in the number of TLS/section when comparing the group treated with α PD-1 alone and the group treated with α PD-1 in combination with α CD40. This suggests

that the ability of α CD40 to induce TLS formation is hampered in the presence of α PD-1. Notably, the increase in TLS/section is significant when comparing α CD40 vs rIgG2a, but it is not significant when comparing the combination group vs rIgG2a (Figure 5d). We have rephrased the text in the results section to clearly describe this data (lines 224-226). Notably, with the single treatment regimen, the number of TLS/section is significantly decreased in the combination group vs α CD40, strongly supporting a role of α PD-1 in dampening α CD40-induced TLS formation (Supplementary Fig. 7e).

figure 7a seems to show smaller size for TLS in combination experiment.

Reply: Although α CD40 therapy enhances TLS formation and results in a larger total TLS area, the area of each individual TLS is very variable and there is no significant difference between the groups (Supplementary Fig. 7b). However, in both α CD40 and α PD1+ α CD40 groups we observe a larger percentage of TLS that are bigger than the mean+SD of the rIgG2a group. This has now been indicated in the revised Supplementary Fig. 7b. We have replaced the combination figure in Supplementary Fig. 7a to not mislead the reader to believe that TLS size is generally reduced when α CD40 is combined with α PD1 therapy.

- Suppl Fig 7 brings an interesting comparison of the effect of anti-PD1/antiCD40 and anti-CTLA4/antiCD40 on TLS induction : it is clear that there is no inhibition in the second combination in contrast to the 1st one despite the induction of resistance to anti-ICI by anti-CD40 in both combinations. Fig 7b misses the p values. Both observations may be discussed. It would have been interesting to compare composition of TLS in B cells and T cells in both situations.

Reply: We thank the reviewer for these observations and suggestions.

As pointed out by the referee, it is clear that α PD1 treatment attenuates the CD40-induced enhancement of TLS formation (Figure 5d and Supplementary Fig. 7e), while α CD40-treatment enhances TLS formation also when administered in combination with aCTLA4-therapy (Supplementary Fig. 7h). This is consistent with the reported role of PD-1 in regulating germinal center B cell survival (PMID 20453843), and has been added to the result and discussion (lines 235-236 and 358-361)

We agree with the reviewer that investigating the T and B cell composition of TLS is of interest, and that changes in the relative ratio between these cell types may potentially alter TLS function. When quantifying the CD3 to B220 ratio in TLS found in mice treated with α CD40, α PD1+ α CD40 or α CTLA4+ α CD40, we observe a tendency towards more T-cells in the α PD1+ α CD40 group as compared to the other two groups (Figure R3). However, this tendency does not reach statistical significance. While this observation is indeed interesting, it needs to be further explored and validated before we can draw any conclusions regarding the mechanisms involved and the importance for TLS function.

In Supplementary Figure 7b, the p-values are not reported because there is no statistical significance among the groups. This is due to the high variability of TLS sizes, which span across a wide range of surface areas. However, it is clear that when α CD40 is administered, the proportion of TLS with a bigger surface area is increased as compared to groups that did not receive α CD40. To clarify, we have added percentages on the graph

in Supplementary Figure 7b, representing the proportion of TLS with a bigger surface area than the mean+SD of the rlgG2a group (threshold indicated by a dotted line in the graph).

Figure R3. CD3 to B220 area ratio within the TLS in GL261-glioma bearing mice that received αCD40 (n=5), αPD1+αCD40 (n=5) or αCTLA-4+αCD40 (n=6). Values: mean±SEM. Statistics: Kruskal-Wallis test.

The new set of data bring a potential mechanism of action of anti-CD40 in glioma bearing mice, suggesting that the detrimental effect of anti-CD40 on response to anti-PD1 is systemic. With this new set of data, this story is somehow disconnected from the story on TLS induction by anti-CD40. Nevertheless, both stories are of interest.

Abstract should be re-written.

Reply: We have scrutinized and improved the abstract (lines 25-36).

We thank the reviewer warmly for the positive comments on our work, the careful reading of our manuscript and the valuable suggestions that have helped us improve the quality of this paper.

Reviewer #4 (Remarks to the Author):

The authors have made significant improvements to their manuscript. From the perspective of this Reviewer, no additional experiments are required. However, several elements of the manuscript should be clarified and discussed in further detail as suggested below:

1. Supplementary Table 1. The authors should clarify whether the human tissue samples were from treatment-naïve patients (i.e. de novo) or were obtained after recurrence and thus, following prior adjuvant therapy. This should also be clarified in the main text on page 8.

Reply: Thanks for bringing to our attention that this important information was missing in the manuscript. All samples included in our cohort were de-novo gliomas collected from treatment-naïve patients. As suggested, this information has been included in the results section (line 163) and in the legend of Supplementary Table 1.

2. Extended data Fig 5. The authors should clarify in the legend when anti-CD40 was delivered.

Reply: Thanks for pointing out this omission. We have clarified the legend as requested.

3. Figure 5d (Line 219). The conclusion that anti-PD1 co-administration impairs anti-CD40-mediated TLS induction is not reflected in Figure 5d as indicated in the text. If by chance there actually is a statistical difference, it is not apparent that this biologically significant (i.e. 0.5 vs 0.3 TLS/section). The increase in TLS with anti-CD40 also seems largely driven by two mice in the anti-CD40 group. The authors should re-consider their conclusion and revise accordingly.

Reply: It is true that we do not see a significant difference in the number of TLS/section between the α CD40 group and the combination group. However, there is also no significant increase in the number of TLS/section when comparing the group treated with α PD-1 alone and the group treated with α PD-1+ α CD40. This suggests that the ability of α CD40 to induce TLS formation is hampered in the presence of α PD-1. Notably, the increase in TLS/section is significant when comparing α CD40 vs rIgG2a, but it is not significant when comparing the combination group vs rIgG2a (Figure 5d). We have rephrased the text in the results section to better describe this data (lines 224-226). Notably, with the single treatment regimen, the number of TLS/section is significantly decreased in the combination group vs α CD40, strongly supporting a role of α PD-1 in dampening α CD40-induced TLS formation (Supplementary Fig. 7e). However, we do agree with the referee that the difference is small and may not be biologically significant, and have therefore not brought this observation forward as a possible mechanism to explain the effect of the combination therapy.

Actually, in Figure 5d, 7 out of 10 mice in the α CD40 group have higher values than any of the mice in the rIgG2a group. To better show the individual values, the graph in Figure 5d has been widened and the symbols indicating data points in the α CD40 group were changed from squares to rhombuses. Notably, a large number of experiments have been performed in this study consistently supporting a role of α CD40 therapy in enhancing TLS formation. Independent data in Figure 1a, Figure 1d, Figure 5d and Supplementary Fig. 7e,h demonstrate that α CD40 therapy increases the number of TLS in glioma-bearing mice.

4. Line 234. The conclusion that the T cells are dysfunctional is a bit misleading. "Reduced functional capacity" may be more appropriate as the T cells are still capable of significant proliferation, activation, and degranulation on ex vivo stim, albeit at modestly reduced levels.

Reply: Thanks for this suggestion on improving our description of T cell functionality. We agree that the terming "reduced functional capacity / reduced functionality" or "T cell hypofunction" are more appropriate and we have revised the manuscript accordingly.

5. **Discussion.** The authors should comment in the discussion regarding their unexpected

findings that anti-CD40 impairs T cell responses in their glioma model. This is important since several other reports have now clearly demonstrated additive or even synergistic activity with combining CD40 agonists with immune checkpoint blockade. Perhaps the authors can further discuss their thoughts on whether this finding might reflect the biological impact of tumors growing in the brain on immune biology in the host and responses to a CD40 agonist. Specifically, prior work has shown that brain tumors can induce sequestration of naïve t cells in the bone marrow of mice and patients which might influence the therapeutic activity of immunotherapy (PMID 30104766).

Reply: We agree that this is an important point since CD40 agonists have been reported to have additive or synergistic effects in non-CNS tumors. One important point is that B-cells are known to be especially important as antigen-presenting cells in the brain (PMID 22028620), which is already mentioned in the discussion. This suggests that the induction of CD11b+ regulatory B cells may be especially deteriorating for immune functions within the brain microenvironment. Important differences in brain tumor immunity as compared to peripheral tumors, including the reference kindly suggested by the referee, has now been added to the discussion (lines 336-344).

6. Discussion. The authors should discuss key limitations to their study. For instance, in addition to the revision where they now indicate in the methods that the glioma cell lines used were of high mutational burden, they should also discuss this in the main text as a potential limitation. Specifically, the cell lines used are of high mutational burden which is not typical of de novo gliomas. In addition, their models are highly responsive to immune checkpoint blockade which is also not characteristic of brain cancers.

Reply: This point is well taken. A paragraph discussing the above-mentioned limitations has been added in the discussion (lines 351-356).

We thank the reviewer for all the valuable input on our study that has helped us improve the work. Your time and effort is much appreciated.

REVIEWERS' COMMENTS

Reviewer #3 (Remarks to the Author):

The authors have answered to all my questions